# miRNA-mediated TUSC3 deficiency enhances UPR and ERAD to promote metastatic potential of NSCLC

Young-Jun Jeon[1,2], Taewan Kim [1], Dongju Park[1], Gerard J. Nuovo[1], Siyeon Rhee [3], Pooja Joshi[1], Bum-Kyu Lee[4], Johan Jeong[5], Sung-suk Suh[6], Jeff E. Grotzke[7], Sung-Hak Kim[8,9], Jieun Song[1], Hosung Sim[1], Yonghwan Kim [10], Yong Peng[1,11], Youngtae Jeong[2], Michela Garofalo[1,12], Nicola Zanesi[13], Jonghwan Kim[4], Guang Liang[14], Ichiro Nakano[15], Peter Cresswell[16], Patrick Nana-Sinkam[17], Ri Cui[1,14] & Carlo M. Croce[1,13]

Non-small cell lung carcinoma (NSCLC) is leading cause of cancer-related deaths in the world. The Tumor Suppressor Candidate 3 (TUSC3) at chromosome 8p22 known to be frequently deleted in cancer is often found to be deleted in advanced stage of solid tumors. However, the role of TUSC3 still remains controversial in lung cancer and context-dependent in several cancers. Here we propose that miR-224/-520c-dependent TUSC3 deficiency enhances the metastatic potential of NSCLC through the alteration of three unfolded protein response pathways and HRD1-dependent ERAD. ATF6α-dependent UPR is enhanced whereas the affinity of HRD1 to its substrates, PERK, IRE1α and p53 is weakened. Consequently, the alteration of UPRs and the suppressed p53-NM23H1/2 pathway by TUSC3 deficiency is ultimately responsible for enhancing metastatic potential of lung cancer. These findings provide mechanistic insight of unrecognized roles of TUSC3 in cancer progression and the oncogenic role of HRD1-dependent ERAD in cancer metastasis.

---

[1] Comprehensive Cancer Center, The Ohio State University, Columbus, OH 43210, USA. [2] Stanford Cancer Institute, Stanford University School of Medicine, Stanford, CA 94305, USA. [3] Department of Biology, Stanford University, Stanford, CA 94305, USA. [4] Institute for Cellular and Molecular Biology, Center for Systems and Synthetic Biology,  The University of Texas at Austin, Austin, TX 78712, USA. [5] Department of Pathology, Stanford University, Stanford, CA 94305, USA. [6] Department of Biosciences, Mokpo National University, Muan 58554, South Korea. [7] Departments of Immunobiology, Yale University School of Medicine, New Haven, CT 06520, USA. [8] Department of Animal Science, College of Agriculture and Life Sciences, Chonnam National University, Gwangju 61186, Korea. [9] Gwangju Center, Korea Basic Science Institute, Gwangju 61186, Korea. [10] Department of Life System, Sookmyung Woman's University, Seoul 140-742, Republic of Korea. [11] Department of Thoracic Surgery, State Key Laboratory of Biotherapy, West China Hospital, Sichuan University, and Collaborative Innovation Center for Biotherapy, 610041 Chengdu, China. [12] Transcriptional Networks in Lung Cancer Group, Cancer Research United Kingdom Manchester Institute, University of Manchester, Manchester M20 4BX, United Kingdom. [13] Department of Cancer Biology and Genetics, The Ohio State University, Columbus, OH 43210, USA. [14] School of Pharmaceutical Sciences, Wenzhou Medical University, Wenzhou, 325035 Zhejiang, China. [15] Department of Neurosurgery UAB Comprehensive Cancer Center, University of Alabama at Birmingham, Birmingham, AL 35294, USA. [16] Departments of Immunobiology, Howard Hughes Medical Institute, Yale University School of Medicine, New Haven, CT 06520, USA. [17] Division of Pulmonary, Allergy, Critical Care and Sleep Medicine, Medical Oncology, The Ohio State University, Columbus, OH 43210, USA. These authors contributed equally: Young-Jun Jeon, Taewan Kim.  Correspondence and requests for materials should be addressed to R.C. (email: ri.cui@osumc.edu) or to C.M.C. (email: carlo.croce@osumc.edu)

Lung cancer is leading cause of cancer deaths in the world. About 80% of all lung cancer are determined as a Non-Small Cell Lung Carcinoma (NSCLC) type. The reasons for the poor overall 5 year survival in lung cancer are multifactorial including late clinical presentation of disease, occult metastatic disease and few targeted therapeutics. Cancer metastasis is a complex, multistep process based in reciprocal interactions between tumor cells and their microenvironment. Although there have been massively studied on this process, the sequence of critical events and molecular mechanisms for cancer metastasis remain poorly understood[1–3].

Cells have Endoplasmic Reticulum (ER) quality control machineries to monitor the proper folding status of a polypeptide. The major response to the accumulation of unfolded and/or misfolded proteins referred to as ER stress is an activation of Unfolded Protein Response (UPR) pathway. There are three commonly described UPR pathways in ER-stressed cells. The contribution of these pathways on tumorigenesis and cancer metastasis are context-dependent and vary based on the duration and strength of the ER stress[4,5]. To date, PERK-eIF2α axis and the IRE1α pathway have been described to enhance and suppress cancer progression in different contexts[4–6]. Additionally, ATF6α-dependent UPR has shown cytoprotective effects leading to oncogenic roles in tumorigenesis. Consistently, several in vitro and in vivo experiments suggest that ER molecular chaperons induced by active ATF6α play critical roles in cancer progression[7–10]. Endoplasmic Reticulum Associated Degradation (ERAD) is a constitutive protein degradation pathway and is highly activated upon ER stress induction. HRD1 protein is an E3 ubiquitin ligase in mammalian ERAD machinery[11]. Although HRD1-specific substrates have been identified, the exact roles of HRD1 in cancer progression are unclear. HRD1 degraded the gp78 protein known as a pro-metastatic ERAD E3 ubiquitin ligase[12,13]. Additionally, the HRD1 has been shown to limit the progression of breast cancers[14]. However, HRD1-dependent ERAD showed anti-apoptotic activity against ER-stress induced cell death[15]. Also, HRD1 protein could promote cancer progression by cytosolic p53 degradation through its E3 ubiquitin ligase activity[13,16,17].

The loss of the chromosomal arm 8p where TUSC3 gene locates is associated with cancer progression and TUSC3 deficiency is frequently observed in advanced stage tumors[18–21]. The roles of TUSC3 in cancer progression showed context-dependent manner in several solid tumors. TUSC3 was initially identified as a homozygous deleted gene in metastasized prostate cancer patients and its deficiency upregulated cancer progression and tumorigenesis in ovarian, prostate, glioblastoma and pancreatic cancers[20–24]. However, it has been also reported that TUSC3 was associated with genetic amplification or enhanced cancer progression in head and neck cancer and colorectal cancer[20,25,26]. In particular, the controversial observations were reported in TUSC3-dependent oncogenesis in lung cancer[27–30]. To date, none of the plausible mechanisms investigated TUSC3-dependent regulation of lung cancer progression and metastasis. miRNAs, the most abundant endogenous small non-coding RNA, are often dysregulated in most cancers, which consequently controls key regulators in tumorigenesis[31]. miR-520c was initially characterized as a pro-metastatic miRNA in breast cancer whereas miR-520c/-373 functional cluster functioned as a metastatic suppressor in estrogen receptor negative breast cancer[32,33]. However, in lung cancer, the function of miR-520c remains unexplored and that of the miR-224 was also controversial in NSCLC tumorigenesis[34,35]. Here we show that miR-224/-520c-mediated TUSC3 suppression enhances the metastatic potential of NSCLC through the altered ER-stress responses and HRD1-dependnet ERAD.

## Results

**TUSC3 deficiency by miR-224/-520c in lung cancer.** To analyze TUSC3 expression in lung cancer, we employed 20 normal, 14 primary and 21 lymph node metastasized lung tumor tissues. TUSC3 mRNA expression using qRT-PCR analysis were found to be reduced in metastatic cancer patient tissues compared to primary and/or normal lung tissues (Supplementary Fig. 1a). The downregulation of TUSC3 protein was further confirmed by immunohistochemistry using 50 sets of the primary and its corresponding metastasized tumor samples (Fig. 1a and Supplementary Fig. 1b). Moreover, TUSC3 downmodulation was associated with poor cancer survival rate ($n = 223$, Fig. 1b), suggesting that TUSC3 could play an important role in NSCLC metastasis.

A target search using in silico tools predicted that miR-224 and miR-520c were likely to target TUSC3 at 102-108bp and 22-28bp downstream of the 3'UTR, respectively (Supplementary Fig. 1c). The luciferase activity using TUSC3 3'UTR was decreased by either miR-224 or miR-520c overexpression but rescued in luciferase activity with 3'UTR deleted in the binding sites (Fig. 1c). Consistently, miR-224 and -520c overexpression suppressed TUSC3 mRNA and protein levels, whereas the suppression of those miRNAs enforced by anti-miRNA inhibitors enhanced TUSC3 expression (Fig. 1d, e and Supplementary Fig. 1d). We next sought to determine if there is an inverse correlation between the miRNAs and TUSC3 gene expression in vivo. We performed an in situ hybridization analysis with 5'-dig-labeled LNA probes for the miRNAs and found that both miR-224 and miR-520c expression was significantly enhanced in metastasized tumor tissues (Fig. 1f and Supplementary Fig. 1e). Moreover, co-expression analyses using IHC assay followed by in situ hybridized samples revealed that there was an inverse correlation between miR-224/-520c and TUSC3 expression in lung cancer patient samples (Fig. 1g, h and Supplementary Fig. 1f, g). These data suggest that miR-224/-520c are responsible for TUSC3 downmodulation and could be used as a biomarker and a therapeutic target for NSCLC. Additionally, TUSC3 proteins were found to be undetectable in 18 of the 50 paired samples, which could be mediated by other mechanisms such as chromosomal deletion or hypermethylation on TUSC3 gene, as well as our proposed mechanism by miR-224 and miR-520c in lung cancer[18,21,27].

**TUSC3 deficiency enhances the metastatic potential of lung cancer.** To understand the roles of TUSC3 downmodulation in NSCLC metastasis, we generated TUSC3 knock-down (KD) H460 and A549 cells using shTUSC3s and found that TUSC3 KD cells showed enhanced migration and invasion. Next, we rescued TUSC3 expression by overexpressing TUSC3 mutants (smTUSC3) harboring silent mutations for the shRNA target sequences and showed a decrease in migration and invasion (Fig. 2a and Supplementary Fig. 2a, 4c). Moreover, the ectopic expression of TUSC3 into HeLa and HCT116 cells, known as TUSC3 null cells, consistently suppressed migration and invasion potential (Supplementary Fig. 3a, b)[36–38]. To identify downstream events in TUSC3 deficient NSCLC cells, we performed PCR arrays with 84 known metastasis-related genes and found that genes involved in cell-to-cell interaction and extra cellular matrix rearrangement were reasonably regulated in both TUSC3 KD H460 cells (Fig. 2b and Supplementary Fig. 3c and Supplementary Dataset 1). These findings are consistent with the previous reports regarding the physiological role of TUSC3 being a member of oligosaccharyltransferase (OST) complex that mediates N-linked protein glycosylation, which frequently involves cell-to-cell and matrix interaction as a

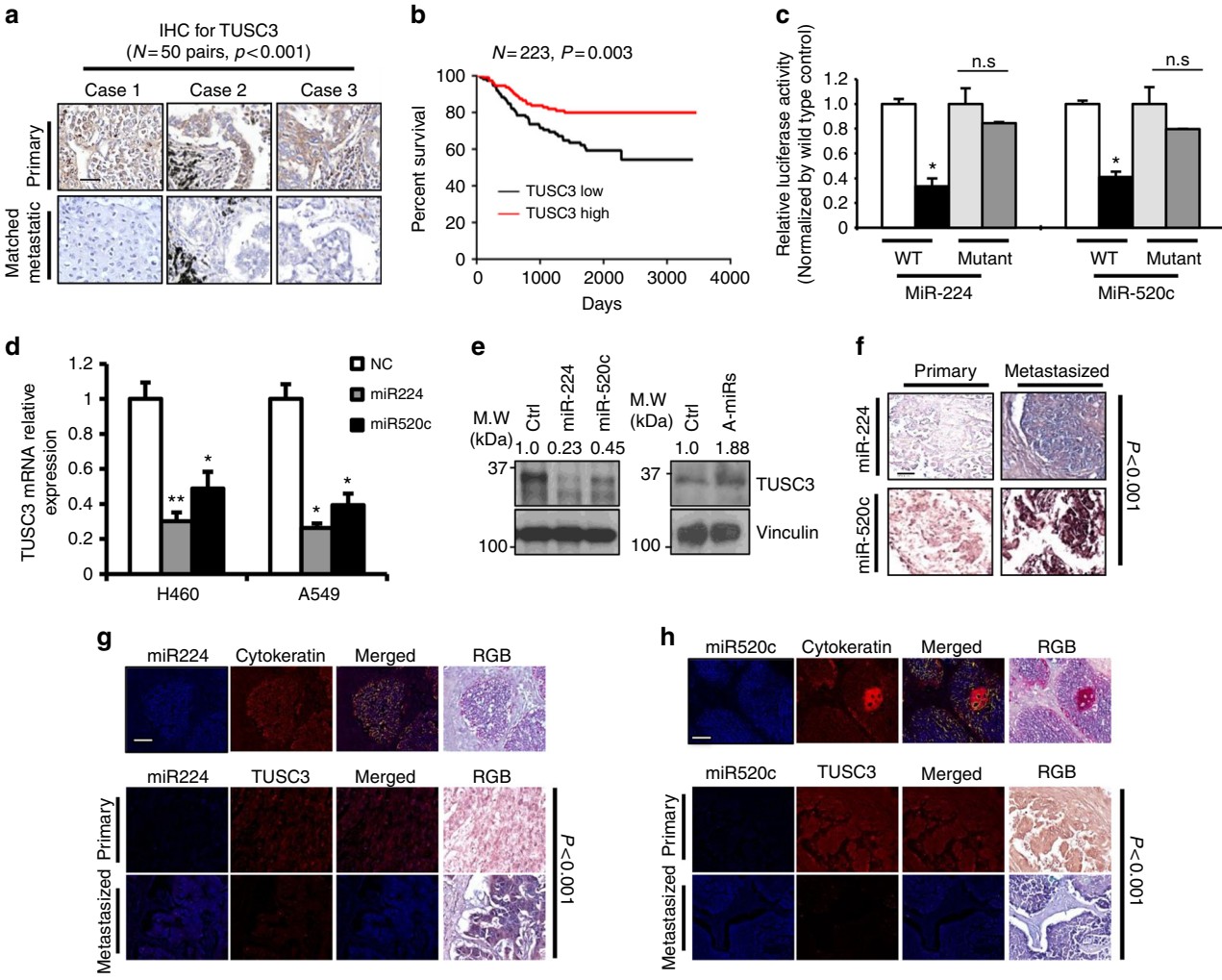

**Fig. 1** miR-224 and -520c are responsible for TUSC3 deficiency in metastasized lung cancer patient tissues. **a** Expression analyses of TUSC3 by immunohistochemistry (IHC) showing suppressed expression in matched metastatic samples. 50 primary and their matched lymph node metastasized lung tumor tissues were analyzed. The summarized expressional scores are shown in Supplementary Fig. 1b. P-value was obtained by Pearson's chi-squared test. **b** PrognoScan based Kaplan Meier plot showing decreased patient survivals associated with TUSC3 deficiency (GSE31210). **c** Luciferase reporter assays with TUSC3 3′UTR in miR-224 or -520c-overexpressing HEK293 cells. The pGL3-TUSC3-3′UTR plasmid or the mutants harboring deleted target sequences for miR-224 or miR-520c were co-transfected with miR-224 (left panels) or miR-520c (right panels). The relative values were obtained by normalizing to Renilla luciferase values. Bars represent means ± SD ($n = 4$) and the p-values were determined by two-tailed student t-test (*$p < 0.001$). **d** Decreased TUSC3 mRNA expression in miR-224 or miR-520c overexpressing NSCLC. Bars indicate means ±SD ($n = 3$) and p-values were addressed by two-tailed student t-test (*$p < 0.01$, **$p < 0.05$). **e** Western blot analyses showing that miR-224 and miR-520c suppressed the endogenous TUSC3 protein in A549 (left panel) and H460 (right panel) cells. Pre-miR-224/-520C (left) or anti-miR-224/-520c (right) were ectopically expressed in A549 (left) or H460 (right) cells. **f** In situ hybridization showing the enhanced miR-224 and miR-520c expression in lymph node metastasized lung tissues compared to their corresponding primary tumors. Total numbers of cases were summarized in Supplementary Fig. 1e. **g**, **h** Co-expression analyses of TUSC3 and miR-224 (**g**) and miR-520c (**h**). The images in the miR-224 and miR-520c sections show a lung adenocarcinoma after co-expression of TUSC3 (fluorescence red and RGB brown) with miR-224 (**g**, fluorescence blue and RGB blue) or miR-520c (**h**, fluorescence blue and RGB blue). The chi-squared test statistic was generated and the null hypotheses that the expression of TUSC-3, miR-224, and miR-520 was equal in primary versus the metastatic tumors or that the expression of TUSC3 and a given miRNA was equal in the primary and metastatic tumors tested using 2 degrees of freedom. The scale bars indicate 150 μm (**a**), 100 μm (**b**), and 200 μm (**g**, **h**), respectively

posttranslational modifier[39–41]. Furthermore, Gene Chip Human Transcriptome Arrays (HTA 2.0) were utilized in A549 TUSC3 knock-out (KO) cells generated by CRISPR KO constructs (Supplementary Fig. 2). After Gene Set Enrichment Analysis (GSEA), metastasis-related gene signatures were shown to be further enriched in TUSC3 KO cells and in Tunicamycin (TM)-treated TUSC3 KO cells (Fig. 2c and Supplementary Dataset 2)[42–44]. These data suggest that TUSC3 downregulation enhances metastasis-related gene signatures and mediates ER-stress induced oncogenesis.

From orthotopic xenograft studies, we confirmed the role of TUSC3 downregulation in promoting cancer metastasis. $5 \times 10^5$ H460 TUSC3 KD cells (sh#2C1) were intravenously injected into nude mice. After 4 weeks, the metastatic nodules from H460 TUSC3 KD cells were increased compared to controls (Supplementary Fig. 3d). Consistently, stronger bioluminescence was detected in the orthotopically xenografted mice injected with A549 GFP[+]/luc[+] TUSC3 KD (#sh3C1-Luc) cells (Fig. 2d). These data strongly suggest that the downmodulation of TUSC3 enhances the metastatic potential of lung cancer in vivo.

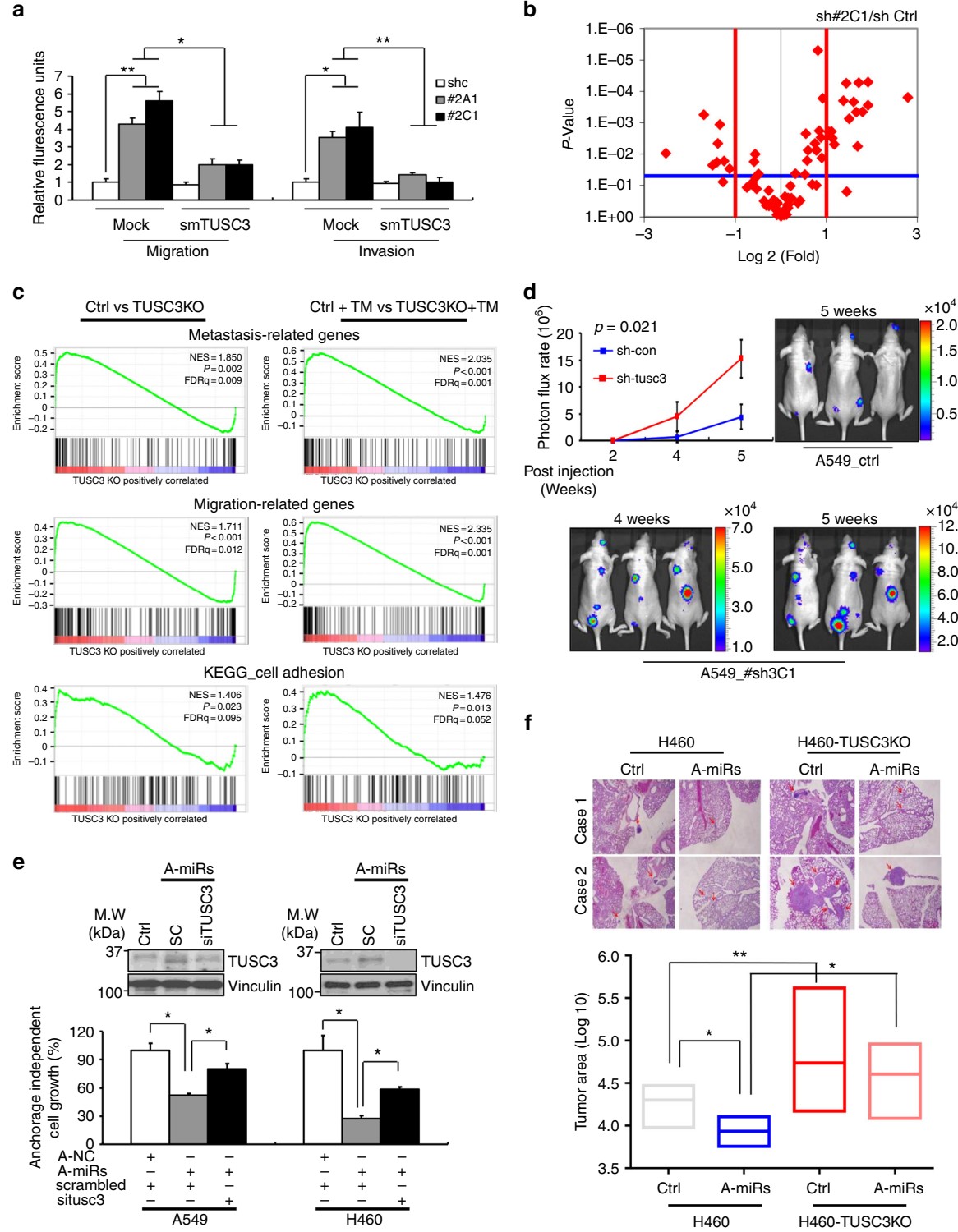

Moreover, miR-224 and miR-520c enhanced cell migratory and invasive abilities (Supplementary Fig 3e, f) and epistatic analyses using anti-miR-224/-520c and siTUSC3 siRNAs consistently showed that the suppression of both miRNAs decreased metastatic potential including anchorage independent cell growth, cell migration and invasion, which was rescued by siTUSC3 siRNAs treatment (Fig. 2e and Supplementary Fig. 3g–3i). Consistently, metastatic potential to lung tissues was suppressed in mice xenografts subcutaneously injected with H460 expressing anti-miR-224/520c miRNAs compared to their corresponding controls, which rescued in H460 TUSC3 KO expressing anti-miR-224/-520c miRNAs (Fig. 2f and Supplementary Fig. 2).

**ATF6α activity is enhanced in TUSC3 deficient cells**. As an ER resident protein, TUSC3 is known as a binding protein with the Oligosaccharytransferase (OST) complex. The studies regarding TUSC3 X-ray structure and biochemical analyses showed that TUSC3 enhanced N-linked glycosylation on the substrates

**Fig. 2** TUSC3 deficiency enhances metastatic potential of NCLC in vitro and in vivo. **a** Increased migration and invasion of TUSC3 knockdown (KD) cells. Two different TUSC3 KD cells (sh#2A1 and sh#2C1) were transfected with empty or mutant TUSC3 (smTUSC3) harboring three silence mutations corresponding to shTUSC3 shRNA target sequences. After 48 h, the cells were introduced into migration (left panels) and invasion (right panels) chambers. Bars represent means ± SD ($n = 3$) and p-value were calculated by student t-test (*$p < 0.05$ and **$p < 0.01$). **b** Differential expression of 84 known metastasis-related genes in H460 TUSC3KD cells (sh#2C1). The lists of genes showing differential expression is shown in Supplementary Dataset 1. **c** Gene set enrichment analysis (GSEA) for metastasis, migration and cell adhesion-related gene sets in A549 control or TUSC3KO cells with DMSO (left panels) or Tunicamycin (TM, 3ug/ml; right panels) for 16 h. The lists of gene sets are shown in Supplementary Dataset 2 (**d**) IVIS in vivo images showing enhanced colonization ability of the A549 TUSC3 KD (sh#3C1). The cells ($1 \times 10^6$) were intravenously injected into nude mice and the bioluminescence was obtained in weekly basis from 3 weeks of injection. Bars indicate means ± SD ($n = 3$) and the p-values were addressed by two-tailed student t-test. **e** Anchorage independent cell growth is modulated by miRNA-dependent TUSC3 downregulation. Anti-miR-224 and -miR-520c were co-transfected with either scrambled or TUSC3 siRNAs for 48 h. After that, the cells were placed into an Agar Matrix Layer for 7 days. The numbers of colonies were quantified with a standard MTS assay. The changed expression of endogenous TUSC3 was analyzed by Western blot analysis (upper panel) or qRT-PCR analysis (Supplementary Fig. 3g). Bars indicate means ± SD ($n = 4$) and p-value were calculated by two-tailed student t-test (*$p < 0.005$ and **$p < 0.001$). **f** Lung metastasis of subcutaneously xenografted mice showing enhanced lung metastasis. The $5 \times 10^5$ of the indicated cells were subcutaneously injected into 4 nude mice, respectively. After 4 weeks, the lung tissues were harvested and dissected in 50-micron depth followed by H&E stain. Tumor areas were measured by Image J software. p-value were calculated by two tailed student t-test (*$p < 0.001$, **$p < 0.005$)

through Cys-X-X-Cys (CXXC) motif on its ER-luminal domain[39–41]. The notion that an abnormal glycosylation in ER is a typical ER stress inducible condition motivated us to analyze UPR activation in TUSC3 deficient cells. Further analysis using GSEA showed that the UPR related genes were highly enriched in TUSC3 KO cells without any stimulation and further enriched in TM-treated TUSC3 KO compared to their corresponding controls (Fig. 3a and Supplementary Dataset 3). However, we could not find any significant difference in the expressional and spatial changes of TUSC3, and the expression of miR-224/-520c in response to ER stress induction (Supplementary Fig. 4). Therefore, we hypothesize that ER stress induction is not a regulator of TUSC3 expression, but rather genetic and/or epigenetic mechanisms could be responsible for TUSC3 deficiency such as the induction of the miR-224/-520c, promoter hypermethylation and/or genetic deletion of chromosome 8p arm[18,38,45].

To elucidate the mechanism underlying TUSC3-dependent UPR regulation, a series of Western blot analyses were performed. Interestingly, the expression of IRE1α and PERK proteins, and their downstream target genes was suppressed in response to ER stress induction in A549 TUSC3KO cells, while the suppression was restored in A549 TUSC3/HRD1 double KO (DKO) cells (Fig. 3b). Conversely, the induction of ER stress enhanced nuclear localized and chromatin-bound ATF6α proteins in TUSC3 KO cells, which failed to be rescued by TUSC3/HRD1 DKO. Consequently, ATF6α promoter activity was enhanced in A549 TUSC3 KO cells (Fig. 3c and Supplementary Fig. 5a–c). The expression of XBP1 known to be induced by active ATF6α protein was consistently increased in A549 TUSC3KO cells, which was not rescued in A549 TUSC3/HRD1 DKO cells. Furthermore, the XBP1 cleavage induced by active IRE1α was suppressed in A549 TUSC3KO cells with ER stress induction, also rescued in A549 TUSC3/HRD1 DKO cells[46] (Supplementary Fig. 5d). Additionally, ER-heat shock proteins, GRP78 and 94 known as an ATF6α responder and promoting cancer progression, were observed to be increased in A549 TUSC3 KO cells (Fig. 3d and Supplementary Fig. 5e, f)[10]. These data suggest that the activation of UPR in TUSC3 deficient cell could be mediated by ATF6α rather than PERK and IRE1α proteins. Moreover, HRD1 protein is responsible for the downmodulation of the IRE1α and PERK proteins, but the activation of ATF6α was regulated in an HRD1-independent manner.

**C-X-X-C motif in TUSC3 is responsible for activating ATF6α.** To analyze the relationship between glycosylating ability and anti-metastatic role of TUSC3, we performed epistasis analyses by employing wild type TUSC3 or TUSC3 CCSS mutant into TUSC3 KO cells (Fig. 3e). The reconstitution of TUSC3 or TUSC3 CCSS mutant in H460 TUSC3KO cells recovered PERK and IRE1α expression whereas the TUSC3-deleted mutant did not show recovered expression (Fig. 3f). More importantly, the reconstitution of the TUSC3 gene in TUSC3 KO cells followed by subcellular fractionation assay showed decreased ATF6α protein activation, while the rescued activation of ATF6α was subtle in TUSC3 CCSS mutant-expressing cells (Fig. 3g). These data suggest that C-X-X-C motif is responsible for TUSC3-mediated ATF6α activation, which is distinct from the TUSC3-dependent PERK and IRE1α regulation. Therefore, we believe that the altered protein N-glycosylation by TUSC3 downmodulation is responsible for the activation of the ATF6α. To analyze whether HRD1-dependent ATF6α activation could mediate the role of TUSC3 in cancer metastasis, we performed a series of rescue experiments. The ATF6α downmodulation enforced by siATF6α siRNAs suppressed the clonogenicity and invasion whereas the other UPR responders, IRE1α and PERK suppression did not affect those abilities in A549 or H460 TUSC3KO cells (Supplementary Fig. 6a–c). The suppression of the metastatic potential mediated by ATF6α in TUSC3 deficiency was further confirmed by orthotopic xenograft mice models showing reduced the numbers and areas of metastatic foci in H460 TUSC3KO cells stably expressing shATF6α (Fig. 3h). Furthermore, IHC for co-expression analysis using anti-TUSC3 and anti-ATF6α antibodies in primary and metastasized lung cancer patient samples showed that enhanced and/or nuclear localized ATF6α protein was highly inversely correlated with TUSC3 expression. Moreover, those proteins were mutually exclusive in the four cases among five cases having both expression of ATF6α and TUSC3 proteins (Fig. 3i). Additionally, the activation of ATF6α was more found to be in metastasized lung cancer patient samples (Supplementary Fig. 6d, e). These data strongly suggest ATF6α has pro-metastatic property by, at least, mediating TUSC3-dependent metastatic regulation. Next, we also analyzed the metastatic potential of HRD1 gene using HRD1KO cells. We observed that the enhanced metastatic potential by TUSC3 KO was rescued by HRD1/TUSC3 DKO in vitro and in vivo (Fig. 3j and Supplementary Fig. 6f, g), suggesting that TUSC3-dependent metastasis is likely to be linked to HRD1-dependent signaling pathways.

**TUSC3 deficiency enhances HRD1-dependent metastatic potential.** To monitor the TUSC3-dependent ERAD efficiency in a live cell, we employed A1AT-NHK-ddVenus GFP mutant[47]. We generated stable cells in A549 TUSC3 KO cells by transduction with the recombinant lenti-A1AT-NHK-ddVenus virus and the accumulated mutant GFP proteins were monitored under

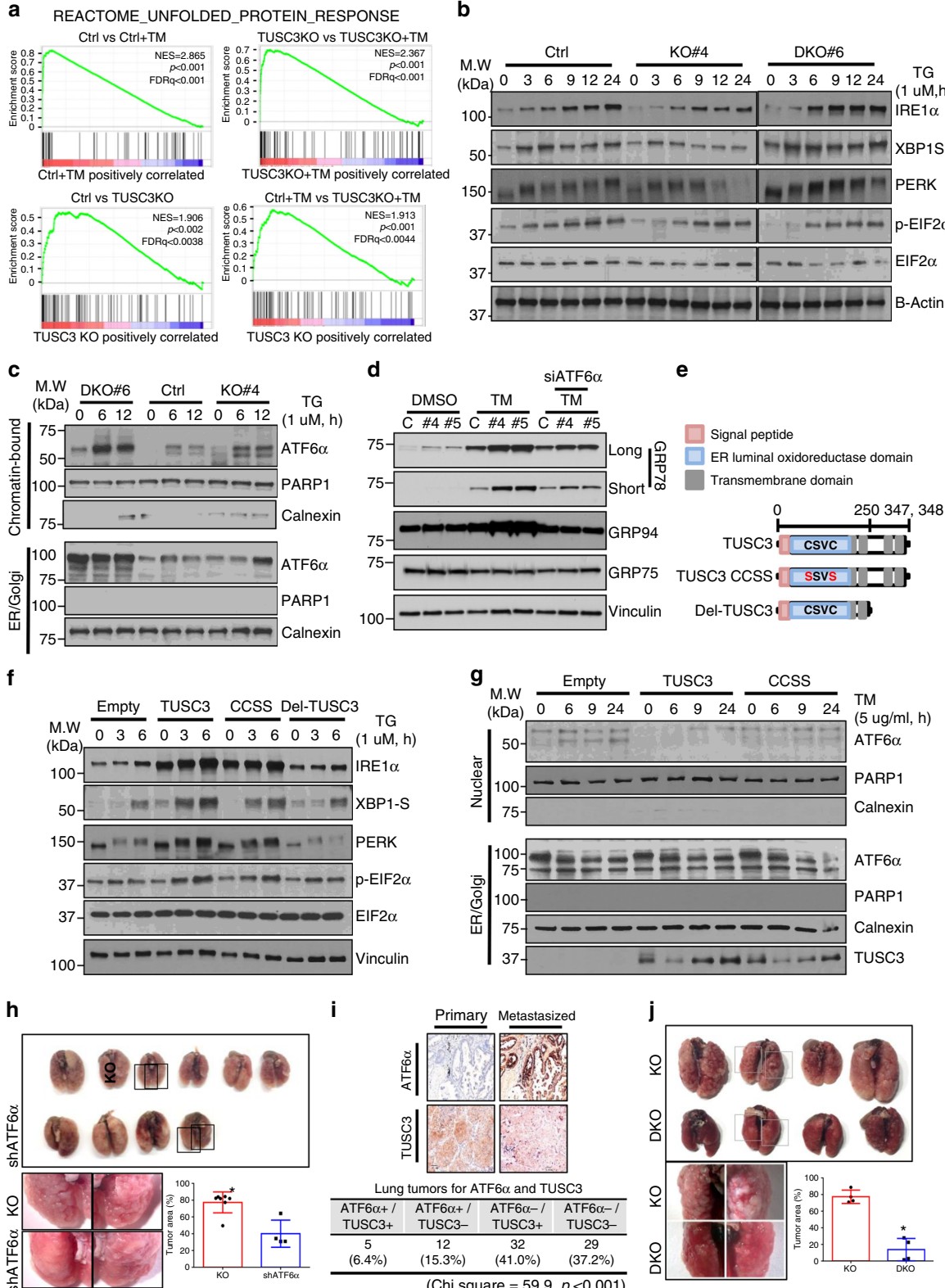

a fluorescence microscope and, quantified by FACS and Western blot analyses. As a result, we found that A1AT-NHK-ddVenus accumulated more in A549 TUSC3 KO cells compared to controls (Fig. 4a and Supplementary Fig. 7a). The accumulation of A1AT-NHK-ddVenus in A549 TUSC3 KO cells were abrogated by overexpression of the TUSC3 gene (Supplementary Fig. 7b, c). Also, anti-miR-224/-520c decreased the accumulation of A1AT-

NHK-ddVenus protein, which was dependent on HRD1 protein (Fig. 4b and Supplementary Fig. 7d). Additionally, stably expressed A1AT-NHK-ddVenus proteins did not accumulated in response to MG132 treatment in both A549 HRD1 KO and TUSC3/HRD1 DKO cells (Supplementary Fig. 7e, f), indicating that the regulation of A1AT-NHK-ddVenus protein by TUSC3 downregulation was completely dependent on HRD1 protein.

**Fig. 3** TUSC3 deficiency selectively modulates Unfolded Protein Responses. **a** GSEA plots showing upregulation of the genes involved in Unfolded Protein Reponses. **b** Western blot analyses showing weakened IRE1α-XBP1 and PERK-EIF2α pathways in A549 TUSC3 KO cells whereas enhanced in TUSC3/HRD1 DKO cells in response to ER stress induction. **c** Subcellular fractionation assay showing enhanced chromatin-bound ATF6α in A549 TUSC3 KO and HRD1/TUSC3 DKO cells. The indicated cells were stimulated by 1.0 uM of TG for 6 or 9 h, respectively. Anti-Calnexin or anti-PARP1 antibody was used for ER/Golgi or Nuclear fraction markers, respectively. **d** Increased ATF6α-dependent ER heat-shock proteins in TUSC3 deficient cells. The cells were transfected by scrambled or siATF6α siRNAs for 48 h followed by exposing to DMSO or TM (3 ug/ml, 16 h). Western blot analysis shows elevated expression of GRP78 and GRP94 in A549 TUSC3 KO cells. A mitochondrial heat-shock protein, GRP75 was used for negative control. **e** Schematic diagram showing primary structure of TUSC3 protein and its mutants (TUSC3 CCSS). CSVC indicates the amino acids in C-X-X-C motif. **f** Rescued IRE1α and PERK expression by reconstitution of TUSC3 or its CCSS mutant. **g** Restored nuclear localization of the ATF6α protein by the reconstitution of TUSC3 but not by CCSS mutant in A549 TUSC3 KO cells. **h** Rescued the colonization ability of H460 TUSC3KO cells by suppressing ATF6α expression. $1 \times 10^6$ of the control cells or ATF6α knock-downed TUSC3 KO ells was intravenously injected into four NOD scid gamma mice. $p$-value was calculated by unpaired student $t$-test ($^*p = 0.002$). The data for the tumor area from the control cells-injected mice are shared with Fig. 5g. **i** Co-expression analysis of ATF6α with TUSC3 protein showing inverse correlation between TUSC3 and ATF6α activation in lung cancer patient samples. $p$-value was obtained by Chi square analysis. The scale bar is shown as 150 μm. **j** Rescued colonization ability of TUSC3/HRD1 DKO cells. The tumor area was calculated as the total area of lung occupied by cancer is field of view using Image J software. The region for the cancer was expressed as percentage. $p$-value was calculated by unpaired student $t$-test ($^*p < 0.001$)

Taken together, these data suggest that TUSC3 deficiency increased HRD1-dependent ERAD activity, mediated by miR-224 and miR-520C.

To examine the direct association of two ER resident proteins, we performed immunoprecipitation (IP) assays using pcDNA-TUSC3-V5-His or its mutants, and pCMV6-HRD1-Flag constructs. We found that the HRD1 protein interacted with wild type TUSC3 and its CCSS mutant but not with the deletion mutant of TUSC3, further confirmed by inverse IP-Western blot analyses (Figs. 3e, 4c and Supplementary Fig. 8a). In addition, a series of pulse-chase experiments with A549 HRD1 KO or TUSC3/HRD1 DKO cells showed that TUSC3 protein levels were not significantly changed in the absence or presence of the HRD1 protein (Supplementary Fig. 8b, c). Consistently, in vivo ubiquitination of TUSC3 protein did not show any significant change in response to HRD1 expression and/or ER stress induction (Supplementary Fig. 8d). Therefore, we believe that the interaction of TUSC3 protein to HRD1 does not affect TUSC3 protein stability.

To characterize the exact role of HRD1-dependent ERAD regulation in TUSC3 deficient cells, we tested HRD1 affinity to two known substrates, p53 and IRE1α[16,17,48]. The endogenous HRD1 protein was precipitated by anti-HRD1 antibody in H460 and H460 TUSC3 KO cells followed by Western blot analyses for HRD1 substrates. As a result, the interactions between HRD1 proteins and its substrates were found to be enhanced by TUSC3 downregulation, which was further confirmed by a rescue experiment that the overexpression of TUSC3 gene in H460 TUSC3KO cells decreased the binding affinity of HRD1 protein to the substrates compared to the KO control. In addition to the known substrates of HRD1, we identified PERK protein as a binding protein with HRD1 and its interactions to HRD1 protein was enhanced upon TUSC3 deficiency. However, another UPR responder, ATF6α was not elucidated in the same immune complex (Fig. 4d).

To establish the competitive binding activity of TUSC3 to HRD1 and p53, we performed an in vitro competition assay with p53 protein and found that its interaction to p53 protein was weakened by TUSC3 protein in HEK293 cells (Fig. 4e). Moreover, poly-ubiquitinated p53 protein was upregulated in A549 TUSC3 KO cells, and rescued upon TUSC3 reconstitution (Fig. 4f). Consistently, p53 ubiquitination was diminished by TUSC3 overexpression and similar results were found with IRE1α (Supplementary Fig. 8e, f). Also, IRE1α and p53 protein was decreased in A549 TUSC3 KO cells, which was slightly rescued by HRD1 deficiency in A549 cells (Fig. 4g and Supplementary Fig. 8g, h). Moreover, the suppressed p53 expression was

recovered by overexpressing TUSC3 or CCSS mutant in H460 TUSC3KO cells treated with DMSO or tunicamycin (Fig. 4h and Supplementary Fig. 8i). In addition, p53-dependent anti-metastatic function was confirmed in orthotopically xenografted mice with H460 TUSC3KO cells overexpressing TP53 gene (Fig. 4i). Furthermore, co-expression analyses of the TUSC3 protein with IRE1α or p53 protein consistently showed that there was a significant correlation between TUSC3 and IRE1α protein and similar results were observed in metastasized lung cancer patient tissues with p53 proteins (Fig. 4j and Supplementary Fig. 8j, k). Also, GSEA plots indicated that genes suppressed by TP53 were enriched in A549 TUSC3KO cells, which was exacerbated in response to tunicamycin treatment (Fig. 4k and Supplementary Dataset 4)[49]. Taken together, TUSC3 downmodulation strengthens HRD1-dependent metastatic potential although biochemical studies should be addressed how TUSC3 regulates the interaction between HRD1 and its substrates.

**TUSC3 deficiency negatively regulates p53-NM23H1 pathway.** To understand the mechanism by which TUSC3 deficiency enhanced metastatic potentials through p53 regulation, we analyzed metastatic suppressor, NM23H1/2[50,51]. First, we observed that the NM23H1/2 proteins were suppressed in TUSC3 deficient cells, which was further suppressed in response to ER stress induction (Fig. 5a and Supplementary Fig. 9a, b). Also, siRNA treatment of the TP53 decreased NM23H1/2 protein (Fig. 5b). Consistently, p53 accumulation by treating with a MDM2 inhibitor, Nutlin3a, recovered expression of NM23H1/2 protein in A549 TUSC3KO cells (Fig. 5c), suggesting that p53 suppression in TUSC3 deficient cells could be responsible for NM23H1/2 suppression. Moreover, rescued NM23H1/2 expression was observed in A549 TUSC3 KD cells treated with siHRD1 siRNAs (Fig. 5d). Next, we sought to determine the involvement of miR-224/-520c in NM23H1/2 regulation and found that the suppression of miR-224/-520c increased NM23H1/2 expression, which was diminished upon TUSC3 reduction (Fig. 5e). Additionally, the ectopic expression of miR-224 or miR-520c decreased NM23H1/2 protein expression levels (Supplementary Fig. 9c, d). According to in silico tools, NM23H1/2 was not predicted as a target for miR-224 or miR-520c (data not shown), suggesting that the alteration of NM23H1/2 expression by miR-224/-520c was indirect effects. Finally, reconstituted NM23H1 gene repressed the migratory and invasive abilities of TUSC3 KD cells, which was further confirmed by orthotopically xenografted mice with NM23H1-overexpressing H460TUSCKO cells (Fig. 5f,g). These data suggest that miR-224/-520c-dependent

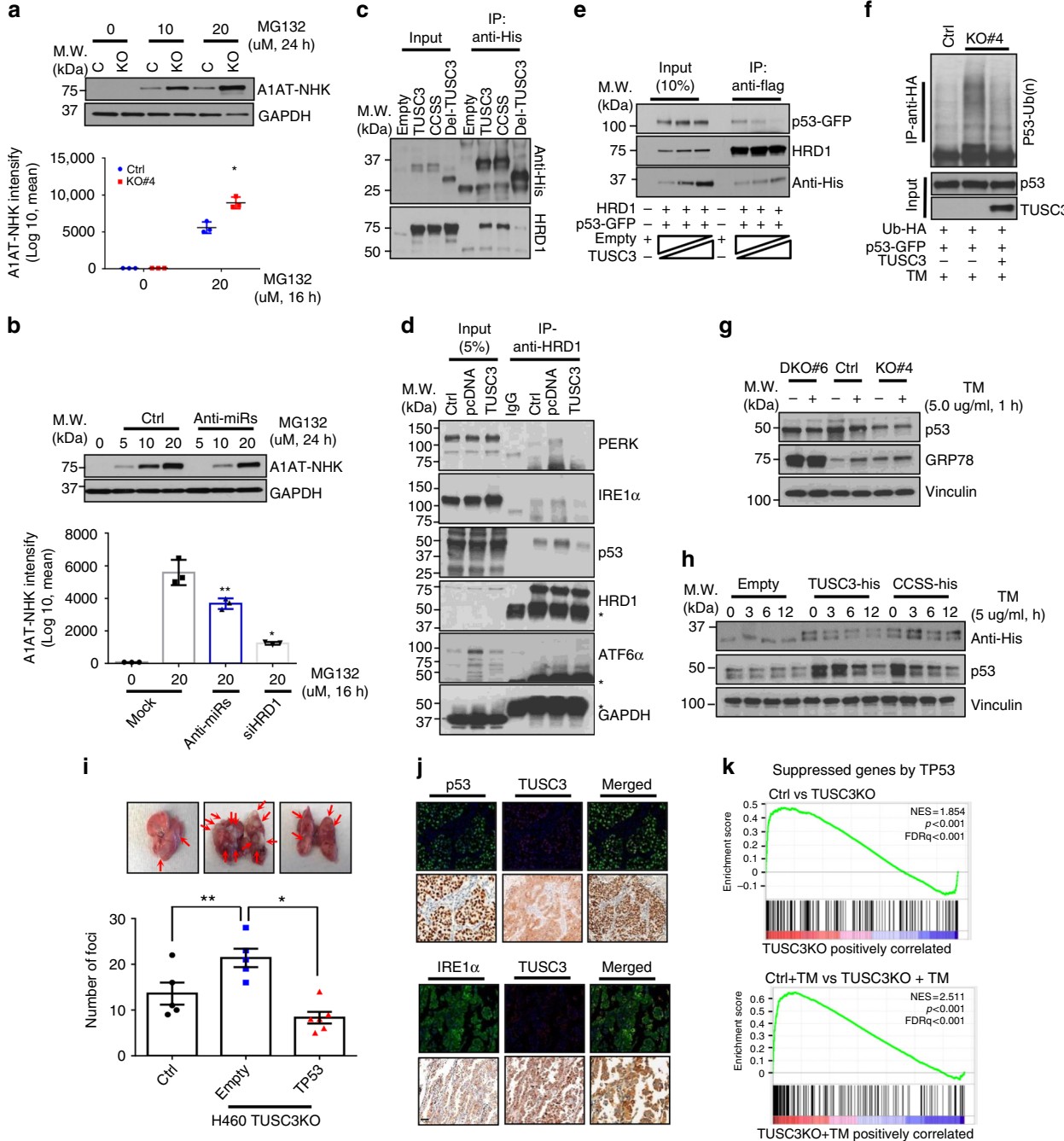

**Fig. 4** TUSC3 deficiency enhances HRD1-dependent pro-metastatic property by enhancing ERAD activity. **a** Enhanced A1AT-NHK-ddVenus GFP accumulation in A549 TUSC3KO cells. The A1AT-NHK-ddVenus stable A549 control (Ctrl) or TUSC3 KO cells were generated by transduction with pLL-Lenti-A1AT-NHK-ddVenus virus. The protein accumulation was monitored after incubating the cells with MG132 (20 uM) for 16 h. The accumulation was quantified by FACS analyses (lower panel) or Western blot analyses using anti-GFP antibody (upper panel). **b** Decreased accumulation of A1AT-NHK-ddVenus by miR-224 and -520c suppression. Bars indicate means ±SD ($n = 3$) and $p$-values were obtained by two-tailed student $t$-test (**a** and **b**, *$p < 0.001$, **$p < 0.02$). **c** Co-immunoprecipitation assays with TUSC3 and HRD1 proteins. pcDNA-TUSC3-V5-His or its mutants (TUSC3-CCSS and Del-TUSC3) was co-transfected with pCMV6-HRD1-Flag in HEK293 cells. **d** The changed affinity of HRD1 protein to its substrates by TUSC3. The endogenous HRD1 protein was precipitated with anti-HRD1 antibody in H460 control, TUSC3 KO or TUSC3-reconstituting TUSC3 KO cells. **e** In vitro competition assay with p53 and TUSC3 proteins in HEK293 cells. **f** In vivo ubiquitination assay of p53 protein showing enhanced p53 ubiquitination in response to TUSC3 downmodulation. **g** Decreased p53 protein in H460 TUSC3 KO cell and rescued in HRD1/TUSC3 DKO cells. The indicated cells were exposed by 5.0 ug/ml of TM for 1 h and subject to Western blot analysis. **h** Rescued p53 protein in TUSC3 or TUSC3-CCSS mutant expressing H460 TUSC3 KO cells. **i** Re-suppressed colonization effect of TUSC3KO cells upon TP53 reconstitution. Tail-vein injection was performed using the indicated cells, and subsequently the lung tissues were harvested after 4 weeks. $p$-values were obtained by student t-test (*$p = 0.00031$, **$p < 0.038$). **j** Co-expression analyses between TUSC3 and p53 (upper panels) or IRE1α (lower panels) in lung cancer patient samples. The p53 and IRE1α are shown in fluorescence green whereas TUSC3 is shown in fluorescence red. Scale bar indicates 150 μm. **k** GSEA plots indicating gene sets suppressed by TP53 were enriched in TUSC3KO cells either DMSO or TM treatment compared to the controls

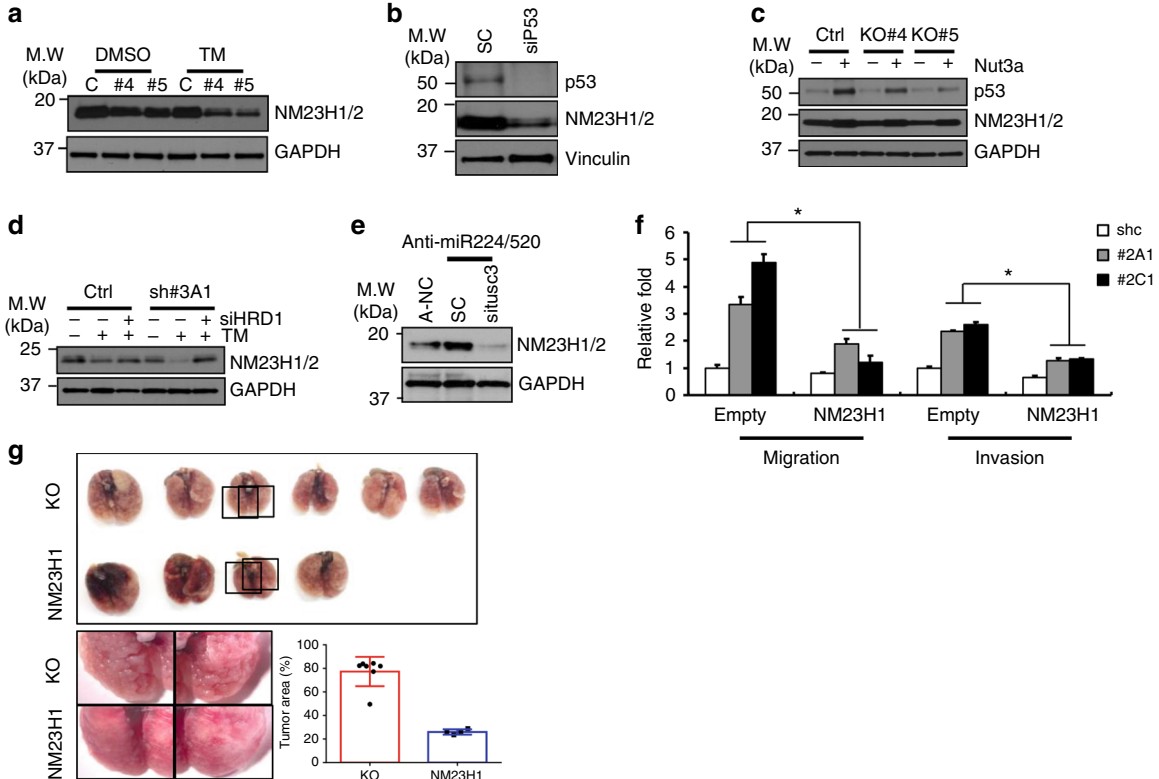

**Fig. 5** Metastatic suppressor, NM23H1/2 is regulated by TUSC3 through p53 regulation. **a** The reduction of NM23H1/2 protein in TUSC3 KO cells in response to ER stress induction. The cells were incubated by 5.0 ug of TM for 9 h and subsequently harvested for Western blot analysis. **b** Decreased NM23H1 protein in p53 deficient cells. The H460 cells were transfected with siTP53 siRNAs for 48 h, and the expression of p53 and NM23H1/2 proteins was measured by Western blot analysis. **c** Restored NM23H1/2 proteins by p53 protein accumulation in A549 TUSC3 KO cells. The cells were incubated with Ntulin3a for 24 h, and Western blot analysis was performed to measure the expression of NM23H1/2. **d** HRD1-regulated NM23H1/2 protein in A549 TUSC3 KD cells. The A549 TUSC3 KD cells were transfected with scrambled or siHRD1 siRNAs. After 48 h, the cells were exposed to TM (5 ug/ml) for 6 h followed by Western blot analysis using indicated antibodies. **e** Accumulated NM23H1/2 proteins upon miR-224/-520c downregulation. The miRNA KD cells were transfected by scrambled or siTUSC3 siRNAs for 48 h, and the NM23H1/2 protein was measured by Western blot analysis. **f** The reduced migration and invasion of A549 TUSC3 KD cells in response to NM23H1 reconstitution. H460 TUSC3 KD cells were transfected by pCMV6-NM23H1 plasmid for 24 h. Subsequently, the cells were harvested and subject to the chambers of migration or invasion assays. Error bars indicate means ± SD ($n = 3$) and the $p$-values were calculated by two-tailed student t-test ($*p < 0.03$). **g** Restored expression of NM23H1 decreased colonization ability of H460 TUSC3KO cells. The cells overexpressing NM23H1 generated by the transduction of lent-NM23H1 virus. $1 \times 10^6$ of NM23H1 overexpressing TUSC3KO cells or control KO cells was intravenously injected into four NSG mice. The tumor area was obtained by Image J software by measuring the field of view in lung occupied by cancer. The region for the cancer was expressed as percentage. $p$-value was calculated by unpaired student t-test ($*p < 0.001$)

TUSC3 suppression regulates NM23H1/2 expression, which regulates the metastatic potential in NSCLC.

Here we present a study characterizing the role of miRNAs-224/-520c-induced TUSC3 downregulation and its contribution to the metastatic potential in NSCLC. In normal or early localized lung tumor, the expression of miR-224/-520c was low while TUSC3 expression was maintained at normal level. As a result, TUSC3 regulates HRD1-dependent ERAD through direct interaction and is involved in N-linked protein glycosylation to some proteins through C-X-X-C motif (Fig. 6, left panel). However, genetic deletion and/or epigenetic modification such as induction of miR-224/520c in advanced lung cancers is responsible for TUSC3 deficiency. As the consequence of aberrant N-glycosylation on some proteins, TUSC3 deficiency enhances the UPR pathways and the E3 ubiquitin ligase activity of HRD1, resulting in the suppression of IRE1α and PERK proteins, which can reduce excessive UPR. Moreover, another HRD1 substrate, p53 is decreased, which weaken the activation of the TP53-NM23H1/2 pathway, which consequently enhances the metastatic potential of NSCLC (Fig. 6, right panel).

## Discussion

An ER-membrane protein, TUSC3 functions in cancer pathogenesis by context dependent manner, but remains controversial in lung cancer. Like most other cancer types, it was initially reported that TUSC3 gene frequently methylated in NSCLC compared to blood lymphocytes[27]. Moreover, the expression of TUSC3 gene was associated with lymph node metastasis and consistently suppressed in advanced lung cancers including small cell lung cancer and lung adenocarcinoma, a subtype of NSCLC[30,52]. Subsequent cell biological assays showed TUSC3 inhibited lung cancer cell proliferation and induced cell death[52]. However, these findings have been challenged by opposite results that the methylation of the TUSC3 gene was associated with increased survival of patients with lung cancer[28]. Additionally, TUSC3 enhanced lung cancer cell proliferation associated with Hedgehog signaling pathway, which was confirmed in tumor xenograft mice model and NSCLC patient samples[29].

Regarding this, our current study claims four novel findings to support the pro-metastatic role of TUSC3 deficiency. First, we observed that TUSC3 expression was frequently suppressed in the

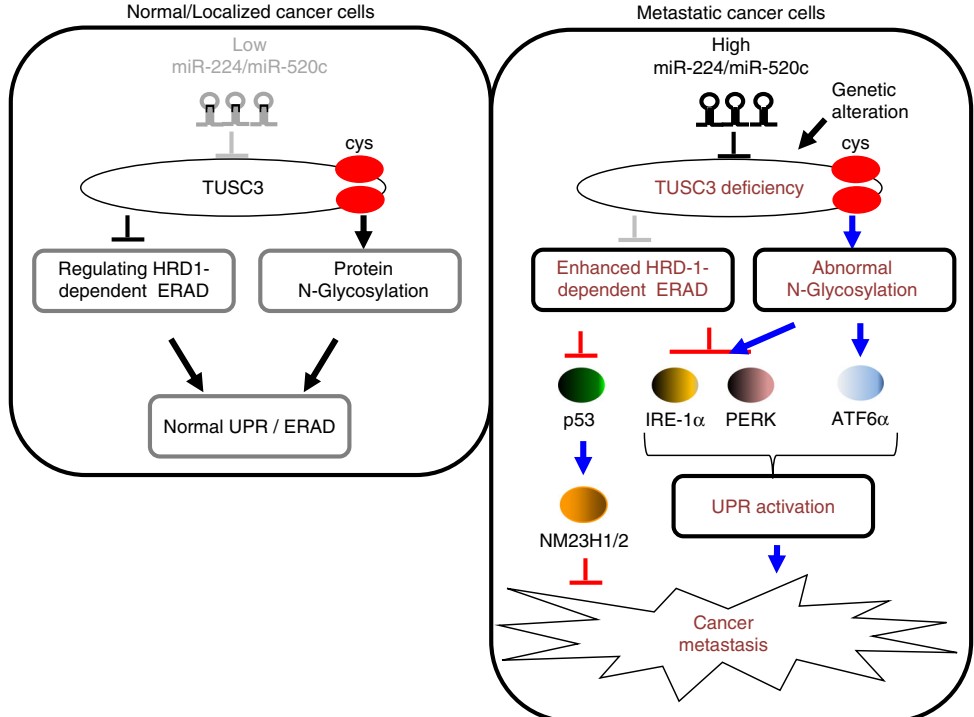

**Fig. 6** The schematic diagram showing the working hypothesis by which miR-224/-520c-induced TUSC3 suppression enhanced metastatic potential of NSCLC through the alteration of UPRs and HRD1-dependent ERAD

metastasized lung cancer patient samples compared to primary lung cancer, possibly mediated by the increased expression of miR-224 and/or miR-520c (Fig. 1 and Supplementary Fig. 1). Additionally, the role of miR-224 also remained controversial and that of miR-520c has not been characterized in lung cancer pathogenesis[34,35]. Our current study could also provide a mechanistic insight to the unrecognized role of the miRNAs. Second, a series of in vitro and in vivo experiments using TUSC3 KD and/or KO cells consistently showed that TUSC3 deficiency enhanced the metastatic potential of NSCLC (Fig. 2 and Supplementary Fig. 2, 3). Third, we also characterized the mechanism underlying TUSC3 deficiency. Utilizing TUSC3KO and TUSC3/HRD1 DKO cells, we found that ATF6α pathway, but not IRE1α and PERK pathways was selectively enhanced in TUSC3 deficient cells, possibly mediated the function of TUSC3 deficiency in lung cancer metastasis. Lastly, TUSC3 regulated HRD1-dependnet ERAD and its deficiency consequently enhanced the HRD1 activity. As a result, the HRD1-dependent ERAD substrates, the PERK, IRE1α and p53 protein were decreased. Moreover, p53-NM23H1/2 tumor suppressive pathway in TUSC3 deficient lung cancer cells was suppressed (Fig. 5 and Supplementary Fig. 9). Interestingly, IHC results also showed that TUSC3 was overexpressed in other stromal cells regardless of the expression of TUSC3 protein in cancer cells, suggesting that the TUSC3-dependent metastatic regulation could be a cell autonomous effect (Fig. 1a).

To date, the significance of UPR activation is not well understood due to their dual roles in terms of cancer progression. One of the accepted principles regarding the role of UPR in cancer is the ability of the UPR to activate tumor suppressive pathways when the cells were under acute and strong UPR activation conditions[4,5]. Our current observation that TUSC3 deficient cells were enriched for gene sets with metastatic signatures compared to control cells led us to investigate the relations between TUSC3 and ER stress response. Although TUSC3 is a known subunit of the OST complex responsible for N-linked protein glycosylation,

the overall impact on N-linked protein glycosylation by TUSC3 downregulation remained subtle because of the functional redundancy of MagT1 protein, another OST associated protein[53]. However, TUSC3 x-ray structure and biochemical analyses revealed that TUSC3 mediated N-linked protein glycosylation through Cys-X-X-Cys (C-X-X-C) motif involving disulfide interactions that makes stable interaction of OST complex to their substrates[39,41]. Moreover, TUSC3 specifically enhanced the efficiency of N-linked glycosylation on integrin β1 and lectin protein family, which enhanced cancer progression in ovarian and prostate cancers[22,54]. Additionally, the alteration of posttranslational modification is frequent ER-stress inducible condition. Indeed, the ATF6α in TUSC3 deficient cells was still activated in HRD1/TUSC3 DKO cells and was only minimally rescued by TUSC3-CCSS mutant overexpression, suggesting that the activation of ATF6α by TUSC3 deficiency is not dependent on HRD1 protein and the function of the substrate stabilization through C-X-X-C motif in TUSC3 is an important for ATF6α activation (Fig. 3e, g and Supplementary Figa. 5a–5c). Therefore, we believe that the upregulation of UPRs in TUSC3 KO cells could be caused by defects in glycome profiles.

We also characterized how three UPRs are independently regulated in TUSC3 deficient cells. A series of rescue experiments using TUSC3 or TUSC3-CCSS mutant gene in TUSC3KO and/or HRD1/TUSC3 DKO cells showed that IRE1α and PERK proteins were downregulated whereas ATF6α was activated in TUSC3 deficient cells with ER stress induction. Importantly, HRD1/TUSC3 DKO cells exhibited restored expression of IRE1α and PERK proteins compared to the TUSC3 KO cells. Furthermore, TUSC3 and its null mutant (TUSC3-CCSS) for N-linked glycosylation can bind to the HRD1 protein. When we reconstituted the TUSC3 or TUSC3-CCSS in TUSC3 KO cells, the PERK and IRE1α proteins were found to be rescued in both TUSC3 and the CCSS mutant expressing TUSC3 KO cells (Figs. 3, 4 and Supplementary Fig. 4). Moreover, IRE1α was known to be a substrate for HRD1-dependent ERAD in synovial fibroblasts and this

regulation was conserved in lung cancer cells, and the PERK, but not ATF6α protein was identified in HRD1 complex (Figs. 3, 4 and Supplementary Fig. 7f)[48]. Consequently, among three UPR responders, PERK and IRE1α were suppressed by the enhanced HRD1-dependent ERAD whereas ATF6α activation is not dependent on HRD1 protein, but rather activated by TUSC3-dependent N-linked glycosylation. We believe that the differential regulation of the three pathways was to protect the cells from excessive UPR activation suppressing tumor progression and metastasis, which makes TUSC3-dependent UPR oncogenic[4,5].

The most prominent dysregulated gene, TP53 was often deleted in lymph node metastasized larynx and pharynx carcinomas where TUSC3 gene was also deleted although there was no any evidence that one gene was deleted along with the other gene[55]. Co-expression analysis using anti-TUSC3 and anti-p53 antibodies showed that there was significant correlation between those in the lymph node metastasized lung cancer patient samples (Fig. 4j and Supplementary Fig. 8j). Moreover, p53 protein was also known to be one of the best characterized substrates in HRD1-dependent ERAD[16,17]. Additionally, HRD1 activity to its substrates was enhanced in response to TUSC3 deficiency (Fig. 4 and Supplementary Fig. 7). Also, the expression of the NM23H1/2 known as a metastatic suppressor was transcriptionally regulated by p53 protein and the suppressed NM23H1/2 expression was observed in our PCR array analyses using H460 TUSC3 KD cells (Fig. 2b, Supplementary Fig. 3c and Supplementary Dataset 1)[51]. These observations motivated us to analyze TP53-NM23H1/2 pathway in TUSC3 deficient cells. Indeed, the p53 protein was regulated by the altered HRD1-mediated ERAD activity in TUSC3KO cells and TP53-NM23H1/2 pathway functionally rescued the metastatic potential in TUSC3 deficient cells in vitro and in vivo (Figs. 4, 5 and Supplementary Fig. 8, 9).

Taken together, we propose that miR-224/-520c dependent TUSC3 downregulation enhances the metastatic potentials of NSCLC by imbalance of the activation of the UPR and enhancing HRD1-dependent p53 protein suppression. Our current observation could provide important insight to understanding the controversial roles of TUSC3 in lung cancer metastasis and the specific roles of UPR and HRD1 regulation in TUSC3-deficient lung cancer, as an anti-cancer therapeutic strategy against NSCLC.

## Methods

**Plasmids and siRNAs**. pCMV6-TUSC3, pCMV6-NM23H1, and pCMV6-Synoviolin (HRD1)-Flag plasmids were purchased from Origene. pcDNA-IRE1α-GFP, pcDNA-TP53-GFP, pcDNA-Ubiquitin-HA plasmids were purchased from Addgene Company. For the TUSC3, the cDNAs were subcloned into pcDNA4.1-V5-His vector with EcoR I and Xho I restriction sites or pcDNA3-HA vector with Hind III restriction site. The 3'UTR of TUSC3-1 and TUSC3-2 plasmids were purchased from Origene and the 3'UTRs were subcloned into pGL3 Vectors with Xho I restriction site. On-Target Smart Pools containing 4 different siRNAs on each target gene of TUSC3, TP53, HRD1, ATF6α or NM23H1 were purchased from Dharmacon company. The primer sets for gene manipulation are shown in Supplementary Tables 1.

**Antibodies**. All dilution factors for the antibodies shown at next of lot number. Otherwise 1:100 dilution of the antibodies from Santacruz Biotech. Inc, or 1:1000 dilution from the other companies were used for the analysis. Anti-FlagM2-resin (#A2220, 1:5000 dilution), anti-GM130 (#G7295) and anti-TUSC3 (#SAB4503183, 1:500 dilution) antibody were purchased from Sigma-Aldrich for IHC and Immunofluorescence assay. Anti-TUSC3 antibodies for Western blot analyses were purchased from (LifeSpan Biosciences, Inc. #LS-C384735; ProteinTech #16039-1, 1:500 dilution). Anti-Calnexin (#9956), anti-CHOP (#9956), anti-phosphor-eIF2α (#3398), anti-GAPDH (#5174), anti-GRP78 (#3177), anti-HRD1 (#14773), anti-IRE1α (#3294), anti-NM23H1/2 (#5353), anti-PARP1(#9532), anti-PERK (#5683), anti-Vinculin (#13901) antibodies were purchased from Cell Signaling Technology. Anti-ATF6α (#sc-22799), anti-eIF2α (#sc-133132), anti-GRP94 (#sc-393402), anti-GRP75 (#sc-13967), anti-HA (#sc-805), anti-P53 (#sc-126), anti-P21 (#sc-397), anti-Ubiquitin (#sc-8017), and anti-XBP1 (#sc-8015) antibodies were purchased from Santacruz Biotech. Inc. Anti-V5 (#R960-25, 1:5000) antibody was purchased from Invitrogen. Anti-His tag (#MCA139. 1:5000 dilution) antibody was purchased from AbD Serotech. Anti-Flag tag (#TA50011, 1:5000 dilution) antibody was

purchased from Origene. Anti-IRE1α (#ab96481, 1:50 dilution) antibody for IHC analysis was purchased from ABcam. Uncropped scans of most important blots are shown in the Supplementary Fig. 10.

**Cell culture and nucleic acid delivery**. The parental TUSC3KO cells, H460 and A549 were purchased from ATCC company. The H460 and A549 (Lung cancer), HeLa (Cervical Carcinoma) and HCT116 (Colorectal Carcinoma) cells were cultured in RPMI 1640 supplemented with 10% fetal bovine serum (FBS) and HEK293 (Human Embryonic Fibroblast) cells were grown in DMEM supplemented with 10% FBS (SIGMA). The transient expression of the plasmids and shRNAs was obtained by using Lipofectamine 2000, Lipofectamine 3000, RNAiMAX, or Lipofectamine Plus reagents according to the manufacturer's protocol (Invitrogen). The contamination of the cell lines used in this study were regularly monitored by a specific kit (Invivogen).

**The generations of TUSC3 knock-down cells**. TUSC3 shRNAs and recombinant lentivirus containing a pool of three target-specific shRNAs of TUSC3 were purchased from Santa Cruz Biotechnology, Inc. For the generation of TUSC3 knock down (KD) cells, H460 or A549 cells were transfected with TUSC3 shRNA plasmids. The transfectants were selected using Puromycin reagent for 2 weeks. After that, the cells were sorted into each single cell and grown for additional 2 weeks. KD candidates were examined for TUSC3 expression by Western blot, qRT-PCR and/or IF analyses. The knock-down efficiencies were confirmed with two independent experiments. The sequences for shRNAs used in this study are shown in Supplementary Table 1.

**The generations of TUSC3 knock-out cells**. The CRISPR knock out (KO) constructs for TUSC3 gene and HRD1 gene were purchased from Santa Cruz Biotechnology, Inc. The cells were transfected with TUSC3 CRISPR/Cas9 KO plasmid. After 48 h, GFP-positive cells were isolated by ARIA FACS sorter and each single cell of TUSC3 KO candidates were plated into 96 well plates. After 10–20 days, the cells were harvested and prepared for Western blot and RT-PCR analyses using anti-TUSC3 antibody, and Taqman probe or Cybergreen PCR primers. For the TUSC3 and HRD1 double KO cells, the TUSC3 KO cells were transfected with CRISPR/Cas9 KO plasmid and HRD1 HDR plasmids for 48 h. After that the cells were selected in Puromycin-containing media for 7 days. The KO candidates were plated into 96 well plates with a single cell. The KO candidates were validated by qRT-PCR analyses of HRD1 Taqman probe and by Western blot analyses with anti-HRD1 antibody. The expression and changed genomic DNA sequences are shown in Supplementary Fig. 2. The information about the sgRNAs and the primers for genomic DNA PCRs to validate knock-out is shown in Supplementary Table 1. The qRT-PCR, Western blot analyses or IF assays to confirm the KO were performed with two or three independent experiments.

**Quantitative RT-PCR**. Total RNA was prepared using TRIZOL according to the manufacturer's protocol (Invitrogen). The 0.5–2 ug of total RNA was subject to first strand synthesis reaction following that thestandard TaqMan miRNA or gene expression assay protocol was performed (Applied Biosystems). The results were produced and monitored in a GeneAmp PCR 9700 Thermocycler. The comparative CT cycles for the quantification of the genes or miRNAs interested was quantified with ABI Prism 7900HT detection system (Applied Biosystems). Subsequently, the values were than normalized by RNU44 and/or RNU48 for the miRNAs and GAPDH, β-actin, and/or OAZ1 for the gene expression, respectively. The sequence information about the probes and/or PCR primers used in current study are shown in Supplementary Table 2. The results of qRT-PCR were performed by three or four technical replicates of two or three independent samples with two independent experiments

**In situ hybridization and co-expression analysis**. Lung cancer tissues were purchased from US Biomax, Inc and the LNA probes for miR-224 and miR-520c were purchased from EXIQON. The in situ hybridization reaction for the miRNAs was done followed by the IHC detection of the TUSC3 on the same section. The section was then analyzed with Nuance computer system which separates each colorimetric base signal as a different fluorescent color, then mixes them to determine if there is co-expression. Optimal detection of TUSC3 by IHC was determined using the Leica Bond Max to be a dilution of 1:100 with a pretreatment in the manufacturer's proteinase K. The optimal detection of miR-224 and miR-520c was determined in proteinase K with a concentration of the digoxigenin-tagged LNA probe of 0.5 pmol/ul. For IHC of TUSC3 protein, each 5 ug/ml of anti-TUSC3 antibody was used. The statistical analysis was performed with the Pearson's chi-squared test by using the InStat Statistical Analysis Software.

**Co-expression analyses**. After using a standard optimization protocol that included positive controls known to have the target of interest, we tested the tissues for the following antigens: TUSC3, p53, IRE1α and ATF6α. Each core in the TMA was scored as positive (at least 20% of the cancer cells showing a strong signal) or negative (less than 20% score). A given tissue was tested for two different antigens using fast red as the chromogen for one target and immunohistochemistry using DAB (brown) as the second chromogen with hematoxylin as the counterstain. The

results were then analyzed by the Nuance and InForm systems in which each chromogenic signal is separated and mixed to determine the percentage of the cells expressing two targets after convered to a fluorescence-based signalt[56,57].

**Migration and invasion assays.** The experimental procedures of migration and invasion assays followed the manufacture's protocols (Calbiochem and TREVIGEN Inc). Briefly, the cells were plated and transfected with indicated plasmids and/or miRNAs, respectively. After 24 h, the cells were incubated with corresponding cell culture media with reduced serum percentage (1%) for additional 12–24 h. Subsequently the cells were washed with PBS and $5 \times 10^5$ or $1 \times 10^6$ cells were plated into the upper chambers of the Migration/Invasion assays in the cell culture media without FBS. The media supplemented with 10% FBS were added into lower chambers to use as a chemoattractant. After 16–24 h for migration assay and 36–48 h for invasion assay, the upper chambers were transferred into a new plate with detaching solutions contained Calcein AM for 0.5–1 h. The fluorescence was analyzed at an excitation wavelength of 485 nm and an emission wavelength of 520nm. The experiments regarding migration and/or invasion abilities with control and/or modified cells were successfully repeated two or three times.

**Xenograft experiments.** Animal experiments were conducted after approval of the Institutional Animal Care and Use Committee, the Ohio State University. The nude mice each were intravenously injected with $5 \times 10^5$ of H460 TUSC3 KD cells or control cells (Jackson laboratory). Four weeks after injection with H460 TUSC3 KD cells, the mice were euthanized, and their lungs were biopsied. The samples were prepared for H&E staining and the number of foci was counted under light microscope (Nikon, Eclipse 50i). For the IVIS in vivo imaging, the stable A549 TUSC3 knockdown and GFP$^+$/luc$^+$ cells were generated, and nude mice were injected (i.v.) with the $1 \times 10^6$ A549 TUSC3 KD and GFP$^+$/luc$^+$ cells or control cells. Luciferase activity was monitored using the IVIS in vivo imaging system weekly for 6 weeks by i.v. injecting in vivo luciferin reagents (Promega). For the in vivo tail-vein injection with TUSC3 KO and HRD1/TUSC3 DKO cells or H460 TUSC3KO, H460 ATF6α KD/TUSC3KO, H460 NM23H1/TUSC3KO cells $(0.1–1 \times 10^6$ cells) were injected intravenously into two different groups consisting of 4-6 NSG mice. After 4 weeks, the colonization ability was obtained by calculating the tumor area in the lung tissue with Image J software. The xenograft experiments were performed once. For the detection of lung cancer metastasis after subcutaneous injections, the H460, H460/anti-miR-224/-520c, H460 TUSC3KO, or H460 TUSC3KO/anti-miR-224/-520c cells were subcutaneously injected into four nude mice, respectively. After 4 weeks, the mice were euthanized, and lung metastases were evaluated under microscope after H&E stain. Specifically, the biopsied lung tissues were dissected with 6 different sections with 50 microns in depth per mice. Tumor areas were measured by Image J software. 6-12 weeks old mice were used in this study and their ages were matched to each group within an experiment and the animals were evenly allocated to each group in an experiment in the case that their genders were different. No statistical analysis was used to predetermine the sample size.

**Recombinant lentiviral production.** A1AT-NHK-ddVenus cDNA was subcloned into pLL-CMV-Puromycin lentiviral vector. The TUSC3 lentivirus was harvested by co-transfection of pLL-TUSC3 and packing constructs in HEK293 cells. After 72 h, the supernatant was collected and enriched for the recombinant viruses by incubating with a virus precipitation solution, PEG-iT$^{tm}$ (System Biosciences) for 16 h at 4 °C. The recombinant lentivirus was recollected by centrifuging the samples at $1500 \times g$ for 45 min at 4 °C.

**Subcellular fractionation assay.** The cells were incubated with DMSO or TM at indicated time and concentration. After that the ER-Golgi and Nuclear fractions were extracted by a subcellular fractionation protocol (ThermoFisher Scientific.). Anti-Calnexin, anti-GM130, and anti-PARP1 antibodies were used for ER, Golgi and Nuclear fraction markers, respectively. Three independent experiments were consistently repeated in TUSC3 KO cells.

**PCR arrays for metastasis regulating genes.** The overall procedure was performed by the manufacturer's instruction (SABiosciences). Briefly, total RNA purified from TUSC3 KD cells or control cells was subject to PCR array plates containing probes for 84 known metastasis regulating genes. The standard qRT-PCR was subsequently performed. Six housekeeping genes were used for normalizing the expression of the genes (GAPDH, β-actin, β-microglobulin, Ribosomal protein large, PO, and HPRT1). The data from the qRT-PCR was analyzed by using RT$^2$ profiler PCR Array Data analysis software (SABiosciences). The control and TUSC3KD cells were triplicated from independent cultured sets and then the PCR array was performed once.

**Soft agar colony formation assay.** Overall procedure was followed according to manufacturer's protocol (Cell Biolabs, Inc.). A base agar matrix layer was prepared in 96 well plate by solidifying at 4 °C for 30 min After that, a top agar matrix layer was mixed with the $0.5 \times 10^6$ cells/ml of the cells and placed into upper layer of the base agar matrix at 4 °C for 30 min Lastly, RPMI media supplemented with 10%

FBS was added and the cells were grown in a standard cell incubator for 4–7 days. s. Subsequently the cell-matrix layer was dissolved by adding a solubilization buffer and the living cells was quantified by a standard protocol of MTS assay (Invittrogne). The anchorage independent cell growth in control and TUSC3KD or miRNA KD cells was performed twice, independently.

**Statistical analysis.** The results were analyzed using ANOVA and/or two tailed student $t$-test. Data are presented as mean ± standard deviation (s.d.) or standard error of the mean (S.E.M.) of two or more independent biological replicates. The statistical method was not used to predetermine sample size. The experiments and outcome assessment were performed in a blinded way. Only $p$-values <0.05 were considered significant.

**Bioinformatics analysis.** Bioinformatics analysis was performed by using these specific programs: Targetscan and Pictar.

## Data availability

The GEO accession number for the data of transcriptome analysis in A549 and A549 TUSC3 KO (KO#4) cells was reported in this paper is GSE76515. The PrognoScan based Kaplan Meier plot shown at Fig. 1b was generated by previous report (GSE31210). All relevant data are available from the authors upon request. A reporting summary for this Article is available as a Supplementary Information file.

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

## Acknowledgements

This work was supported by grants from the National Cancer Institute (1R35CA197706-01) to C.M.C. and National Natural Science Foundation of China (81672305) to R.C.

## Author contributions

C.M.C. guided the project. Y.J.J. conceived the idea and wrote manuscript. Y.J.J, T.K., and R.C. designed the experiments. YJ.J. and R.C. designed and performed in vivo mice studies. G.N. performed in situ hybridization and immunohistochemistry experiments. Y.J.J., T.K., P.J., D.P., R.C., S.R., J.J., H.S., and Y.P. performed in vitro experiments. Y.J.J, T.K., I.N., Y.J., and S.K. analyzed cancer patient samples. Y.J.J., R.C., N.J., S.R. Y.J., and S. S. performed mice xenograft experiments and analyses. Y.J.J., B.L., and J.K. conducted GSEA analyses. Y.J.J., J.E.G., and P.C. established the system for ERAD detection. Y.J.J.,T. K., R.C., Y.K., and M.G. analyzed and interpreted data. G.L., P.N., and C.M.C. pre-revised manuscript.

## Additional information

**Competing interests:** The authors declare no competing interests.

