## [Peer Review File · Nature Communications]

This manuscript has been previously reviewed at another journal that is not operating a transparent peer review scheme. This document only contains reviewer comments and rebuttal letters for versions considered at Nature Communications. Mentions of the other journal have been redacted.

Reviewers' comments:

Reviewer #1 (Remarks to the Author):

The manuscript entitles “miRNA mediated TUSC3 deficiency enhances UPR and ERAD to promote metastatic potential of NSCLC” describes an interesting pro-metastatic mechanism were TUSC3 downregulation by miR-224/-520c increase ATF6 activation and ERAD induction in a HRD1 -dependent manner. Authors provide some consistent evidence indicating that TUSC3 expression is regulated and inversely correlated with miR-224/-520c in metastatic cells, however in the revised version of the manuscript most of data about the mechanism of the impact of TUSC3 expression in metastasis is still correlative. The paper do not have the standard yet for Nature Com, but could be clearly improved.

General Comments

As we state before the results described in this manuscript are interesting; however, more evidence is needed to clarify the mechanism of the role of TUSC3 in metastasis. The conclusions then are still overstated and based on too simplistic correlations. We still think that are too many correlations between multiples pathways and more clear and direct experiments are needed to really address the hypothesis.

Specifics Comments

Major concerns:

1. In the first revision of the manuscript we point out that this is not clear in the manuscript what happen with IRE1a, PERK and ATF6 branches in TUSC3 KO cells. The authors conclude that the enhanced HRD1-dependent ERAD by TUSC3 deficiency is responsible for the downregulation of PERK and IRE1 α , whereas ATF6 α is activated. However, the authors don't evaluate PERK and IRE1a activation. Also in Figure 3C there is a higher amount of ATF6 in the ER fraction in the DKO cells, so this means that ATF6 may be also being degraded by HRD-1? The authors should include ATF6 in Figure 3B. Also, it is important include experiments to visualize UPR activation (not only ATF6) in TUSC3 deficient cells, to properly conclude that only the ATF6 is activated by the downregulation of TUSC3. In this analysis, it is important verify the expression of genes targets of the three UPR branches.

Response) As reviewer pointed out, we agree that it is important to show the activation markers of PERK and IRE1 α in TUSC3

deficiency. In general, the regulation and the downstream molecules of UPRs have often shared each other. Thus, we decided to analyze direct target molecules of PERK and IRE1 α activations. We analyzed the phosphorylated EIF2 α for PERK activation

Fig. 1a for Reviewer showing decreased PERK-EIF2 α and IRE1 α -XBP1 activation in TUSC3 deficient cells, but not in HRD1/TUSC3 DKO cells.

and the amount of spliced XPB1 using Western blot analyses, and the induction and cleavage of XPB1 mRNA as an activation marker for ATF6 α and IRE1 α , respectively^{1,2}. As shown in revised Fig. 3b or Fig.

1a for Reviewer, we consistently observed the spliced XBP1 protein and EIF2 α phosphorylation were decreased in TUSC3 deficient cells whereas their expression was rehabilitated in HRD1/TUSC3 DKO

Fig. 1b for Reviewer showing the decreased efficiency of XBP1 splicing in both TUSC3 KO and DKO cells

cells. Moreover, XBP1 mRNA is known to be increased by active ATF6 α and cleaved by active IRE1 α protein¹.

Consistently, we found the cleaved XBP1 mRNA was found to be decreased in A549 TUSC3 KO cells, which was rescued in A549 HRD1/TUSC3 DKO cells in response to ER stress induction (Supplementary Fig. 5d or Fig. 1b for Reviewer). Additionally, the expression of XBP1 mRNA was enhanced in A549 TUSC3KO or HRD1/TUSC3 DKO cells compared to control cells

(Supplementary Fig. 5d or Fig. 1b for Reviewer), which is

consistent with our observation that another marker for ATF6 α activation, GRP78/Bip protein was increased in TUSC3 KO cells (Fig. 3d, Supplementary Fig. 5c). Furthermore, the activations of the target

Previous Fig. 3f

Fig. 2 for Reviewer showing recovered activation of PERK-EIF2 α and IRE1 α -XBP1 in TUSC3 or CCSS-overexpressing TUSC3KO cells.

Revised Fig. 3f

molecules were further confirmed in TUSC3- or CCSS-TUSC3-

overexpressing TUSC3KO cells. As shown in Fig. 3f or Fig. 2 for Reviewer, Western blot analysis showed that the spliced XBP1 and the phosphorylated EIF2 α proteins were rescued upon the reconstitution of TUSC3 or CCSS-TUSC3 mutant in H460 TUSC3KO cells (Fig. 3f or Fig.2 for Reviewer).

As the reviewer's comment, we also recognized that amount of ATF6 α protein was increased in HRD1/TUSC3 DKO cells, suggesting ATF6 α is likely to be also directly regulated by HRD1-dependent ERAD as well. However, we could not find any evidence for ATF6 α protein in the HRD1 complex. Specifically, the endogenous

immunoprecipitation (IP) and Western blot analysis showed there was no interaction between HRD1 and ATF6 α protein (Revised Fig. 4d or Fig. 3 for Reviewer), also confirmed by the overexpression IP in HEK293 cells using ATF6 α and HRD1 constructs (data not shown), suggesting that the regulation of ATF6 α by HRD1 protein could be an indirect effect, which could be another interesting question how HRD1 can regulate the expression of ATF6 α protein. Furthermore, this observation could provide another evidence for our hypothesis that the regulatory mechanism of PERK and IRE1 α is

Previous Fig. 4d

Revised Fig. 4d

Fig. 3 for Reviewer showing the interaction between HRD1 protein and its substrates but not with ATF6 α protein

different from that of ATF6 α in TUSC3-dependent ER-stress responses.

To directly compare the regulation of three responders for UPR in TUSC3 deficient cells, we wish to show ATF6 α activation data in Fig. 3b as the reviewer's suggestion. However, although several antibodies could detect full length of ATF6 α protein, we could not find any proper antibody to detect the activation form of ATF6 α in whole cell lysate because there were strong multiple non-specific and irrelevant bands unresponsive by ER stress stimulation between 40-75 kDa. **That was why we** initially performed the cell fractionation assay to detect the active and nuclear-localized ATF6 α protein, which could provide more direct evidence for ATF6 α activation. Instead, we consistently showed that the activation of the ATF6 α in TUSC3 deficiency was confirmed by several Western blot analyses, ATF6 α promoter assays and analyzing downstream target genes of ATF6 α such as GRP78 and XBP1 inductions (Fig. 3c, 3d, 3g, Supplementary Fig. S5a-5f).

Taken together, our hypothesis that TUSC3 deficiency differentially regulated PERK and IRE1 α proteins from ATF6 α protein could be supported by following observations. First, the expression of PERK and IRE1 α was decreased in TUSC3 deficient cells and their down-stream target molecules such as phosphorylated EIF2 α and XBP1 splicing were consistently regulated in TUSC3 deficient cells but were rescued in the HRD1/TUSC3 DKO cells and in TUSC3 overexpressing TUSC3KO cells. Second, we also provide a mechanistic evidence that PERK and IRE1 α are a HRD1-dependent ERAD substrate. On the other hand, cell fractionation assay, Western blot analysis, confocal imaging analysis and ATF6 α promoter assay consistently showed that ATF6 α is activated in TUSC3 deficient cells. Importantly, the nuclear localized active ATF6 α was not rescued by CCSS TUSC3 overexpression in TUSC3 KO cells, suggesting that ATF6 α is likely to be regulated by the activity of N-linked glycosylation on some proteins by TUC3 protein. Lastly, we also provided the functional relevance of ATF6 α , but not PERK and IRE1 α to mediate the metastatic potential of TUSC3 deficiency (Please see below for our response to next comment).

Regarding our responses to this comment, we replaced previous Fig. 3b, 3f and 4d to the revised ones and added new data in Supplementary Fig. S5d.

2. One of the mechanism proposed in the paper that its involve in the pro-metastatic potential of TUSC3 deficiency it's the activation of ATF6 branch, however there are no experiment provided corroborating this statement. Is there an increase in expression of ATF6 in metastatic cancer patient tissues compared to primary and/or normal lung tissues? Does the downregulation of ATF6 inhibit the metastatic capacity of TUSC3KO cells? The authors should address these question to conclude that ATF6 activation is important for the increased metastatic potential of TUSC3KO cells.

Response) To directly address the question raised by the reviewer, we employed primary lung cancer and their corresponding lymph node metastasized tumor samples to perform the co-expression analysis using immunohistochemistry with anti-ATF6 α and anti-TUSC3 antibodies. As a result, we observed there was a strong inverse correlation between TUSC3 and ATF6 α expression. Interestingly, four of total five cases having both TUSC3 and ATF6 α expression were mutually exclusive. Moreover, the enhanced expression and/or nuclear localized ATF6 α were found to be in the lymph node metastasized patients samples compared to primary lung tumor tissue. (New Fig.3i and Supplementary Fig. 6df,e or Fig. 4 for

Fig. 4 for Reviewer showing that inverse correlation between TUSC3 and ATF6α protein (left), and enhanced expression and/or nuclear-localization in metastasized lung tissue samples.

TUSC3KO cells whereas those treated by siIRE1α or siPERK siRNAs remained unchanged (Supplementary Fig. 6a-6c or Fig. 5 for Reviewer). Also, we observed the suppressed metastatic potential in the tail-vein injected NSG mice with ATF6α stable knock-down TUSC3KO cells compared to the control TUSC3KO cells (Fig. 3h or Fig. 6 for Reviewer). Therefore, we claim that ATF6α mediates the metastatic potential of lung cancer with TUSC3 deficiency.

Regarding this, we added new data in Fig. 3h and 3i, and Supplementary Fig. 6a-6c.

New Supplementary Fig. S6a-S6c

Fig. 5 for Reviewer showing that the suppression of ATF6α expression decreased the invasion (a) and clonogenic abilities (b,c) of TUSC3-deficient cells.

Fig. 6 for Reviewer showing that the decreased metastatic potential of H460 TUSC3KO cells with suppression of ATF6α expression.

3. Is p53 the only HDR-1 target that may have an impact for cancer metastasis? How did the authors chose p53 as the HDR-1- dependent mechanism? The authors should explain more in detail this mechanism and how the study was conducted to these genes. Also, there is no in vivo evidence demonstrating that p53 and NM23H1 mediate the role of TUSC3 in NSCLC metastasis.

Response) As described in our original manuscript, our observation that TUSC3 deficiency enhanced HRD1-dependent ERAD activity initially motivated us to analyze the roles of HRD1 protein in lung

Reviewer). Moreover, we also addressed whether ATF6α could mediate the metastatic potential of lung cancer with TUSC3 deficiency *in vitro* and *in vivo*. We performed the clonogenic and invasion assays in A549 and H460 TUSC3KO cells having suppressed expression of IRE1α, PERK or ATF6α enforced by their specific siRNAs. We consistently found that the suppression of ATF6α expression decreased the clonogenicity and invasive ability in both A549 and H460

cancer tumorigenesis and metastasis. Moreover, the exact functions of HRD1 in cancer progression is not well understood because bona fide functions of its substrates have opposite directionality in cancer progression. Specifically, the other mammalian ERAD-specific E3 ubiquitin ligase, gp78 known as a prometastatic protein is the substrate of HRD1 protein to antagonize it, which consequently suppressed the progression of breast cancer²⁻⁴. On the other hand, ERAD mediated by HRD1 protein enhanced cell survival in ER-stress induced cell death⁵. Also, tumor suppressor p53 protein is degraded through HRD1-dependent ERAD^{6,7}. Therefore, we decided to directly investigate the role of HRD1 in TUSC3 deficient cells. A series of *in vitro* and *in vivo* analyses using HRD1 deficient cells, we found the suppression of HRD1 expression decreased the metastatic potential of TUSC3-deficient cells (Fig. 3j and Supplementary Fig. 6f and 6g). In addition to our data, we were eventually interested in the role of p53 by previous observation that the most prominent dysregulated gene, TP53 and TUSC3 were often deleted in lymph node metastasized larynx and pharynx carcinomas although they did not provide any further data whether these genes were correlated⁸. Furthermore, p53 protein is well characterized substrate for HRD1-dependent ERAD^{6,7}. Those previous observations and our current findings led us to analyze direct relationship between p53 and TUSC3 proteins. Regarding this comment, we described more specifically in the discussion section (page 13, paragraph 2).

ERAD is a constitutive degradation pathway, which plays an important role to maintain ER homeostasis in response to ER stress induction⁹. Considering its importance and diverse responses, the ERAD should be tightly regulated and well organized in cell homeostasis. Moreover, the failure of proper responses for metabolic stresses is often involved in disease progression including cancer. Thus, we are expecting that there should be more substrates and regulating mechanisms to HRD1 protein. It is evident that we confirmed IRE1 α and identified PERK protein as other substrates for HRD1-dependent ERAD although their contributions on cancer progression need to be addressed in other context rather than TUSC3 deficiency. Regarding this, it should be interesting topic identifying a novel substrate for HRD1-and/or gp78-dependent ERAD to characterize how these ERAD-specific E3 ubiquitin ligases cooperatively work in cancer progression. Therefore, we do not think p53 protein is only one major substrate in TUSC3 deficient cells through HRD1 protein in cancer metastasis, which should be also further investigated.

As the reviewer's suggestion, we also conducted a rescue experiment in H460TUSC3KO cells to

Fig. 7 for Reviewer showing that the decreased metastatic potential of H460 TUSC3KO cells overexpressing NM23H1 gene

analyze whether NM23 gene mediates the metastatic potential in TUSC3 deficiency. We established NM23-H1-overexpressing TUSC3KO cells using recombinant lenti-NM23H1-virus and we performed orthotopic xenografting experiment. We observed that NM23-H1 overexpression decreased the areas of the metastatic nodules (Fig. 5g. or Fig. 7 for Reviewer). Moreover, we also showed *in vivo* evidence for the involvement of p53 gene in the metastatic potential in TUSC3 deficiency, which further confirmed in lymph node metastasized cancer patient samples using IHC analysis (Fig. 4i, j and Supplementary Fig. 8j).

Minors comments:

We would like to appreciate these detailed comments, greatly helpful for enhancing the quality of manuscript and figures as well as major comments.

1. In line 62 the authors should change the word caner by cancer.

Response) We fixed this error.

2. In line 104 delete the word shRNAs, it's enough with shTUSC3.

Response) We deleted it.

3. In line 137 correct the word recued by rescued.

Response) We changed it.

4. In line 279 there are a mistake in the number of the figures, please correct this.

Response) As the reviewer pointed out, the order of the figure was wrong, and we swapped the figures 5a and b to fix this error. The authors especially thank you for this correction and apologize for the confusion.

5. Still more information is required to understand the graphs. Figure 1A and 1B should say TUSC3, Figure 1C should say the conditions of each bar, Figure 1D should say TUSC3 mRNA relative expression in the Y axis.

Response) Thank you for the detailed instruction. We changed them as the reviewer's recommendation.

6. There is a mistake in Figure 3C, the figures for PARP1 and Calnexin are probably exchanged.

Response) We would like to appreciate this critical comment. We changed it.

Reviewer #2 (Remarks to the Author):

The authors have made extensive efforts to address the reviewer comments. Most points have been adequately addressed. I would recommend publishing this manuscript if the authors can address the following issues:

1. The authors showed that TUSC3 is downregulated by miR-224/520C. They also mentioned that TUSC3 is frequently hypermethylated in NSCLC. Which mechanism plays a major role in downregulating TUSC3 in NSCLC metastasis?

Response) As the reviewer pointed out, it has been reported in other cancer that there is genomic deletion of chromosome 8p22 locus where TUSC3 gene located or hypermethylation on the promoter region of TUSC3 gene¹⁰⁻¹². Along with these previous literatures, our IHC result using anti-TUSC3 antibody showed that TUSC3 was undetectable in 18 of total 50 pairs in both primary and metastasized lung cancer patient samples (Fig. 1a and Supplementary Fig. 1b). Thus, we initially described that these null expressions could be caused by genetic deletion and/or hypermethylation on TUSC3 promoter. However, we currently do not have any evidence for the null expression of TUSC3 in the 18 pairs. Rather, whether genetic deletion of TUSC3 or 8p22 region has not been reported in lung cancer. Moreover, it was reported that the hypermethylation on TUSC3 promoter was associated with lung cancer progression, but this was challenged by other observation¹²⁻¹⁴. Therefore, our initial description in the original manuscript should be changed by more clear way.

Furthermore, we showed that TUSC3 was found to be suppressed in metastasized lung cancer with 25 of the remaining 38 pairs (over 65% of total case) whereas the miR-224/-520c were highly increased in metastasized lung tissue. Additionally, there was strong inverse correlation between the expression of TUSC3 and miR-224/-520c in both primary and metastasized lung cancer tissues (Fig. 1f-h and Supplementary Fig. 1e-g). Moreover, we also found that miR-224 and -520c directly suppressed TUSC3 expression at mRNA levels (Fig 1c-e and Supplementary Fig. 1c,d). Therefore, we believe that miR-224/-520c-dependent TSUC3 deficiency could be a major mechanism in, at least, lung cancer to evoke or mediate lung cancer metastasis. We appreciate this comment and apologize for the confusion.

To make this clearer, we removed the previous sentence “Additionally, TUSC3 proteins were found to be undetectable in 18 sets of the 50 paired samples, which is consistent with previous observation that the TUSC3 gene was frequently hypermethylated in NSCLC”, and changed our description as shown on page 5, paragraph 1.

2. Considering that TUSC3 regulates p53, is TUSC3 a general lung tumor suppressor or does it specifically regulate lung cancer metastasis?

Response) As the reviewer’s expectation, we also initially hypothesized that TUSC3 could play a general tumor suppressive role in lung cancer progression because the TP53 gene is one of the most frequently mutated genes associated with poor survival of patient with NSCLC and is known to be a strong tumor suppressor¹⁵⁻¹⁷. Moreover, our current observations are claiming that the p53 mediates the metastatic potential of TUSC3 deficiency (Fig. 4i, 4j, 4k, Fig. 5 and Supplementary Fig. S8j, S8k) and characterized the mechanism underlying enhanced HRD1-dependent ERAD activity in TUSC3 deficient cells *in vitro* and *in vivo* (Fig. 4a-4i and Supplementary Fig. S7, S8e, S8g, S8i). Also, it has been known that TUSC3 suppressed cell proliferation in prostate and pancreatic cancer^{18,19}. However, we could not see any significant difference in cell cycle progression and cell survival against ER stress-induced cell death induced by tunicamycin or thapsigargin in the TUSC3 deficient lung cancer cells compared to corresponding control cells (data not shown). Moreover, our initial qRT-PCR analysis using primary lung cancer patient samples showed that there is no any significant change in TUSC3 mRNA expression compared to normal healthy lung tissue (Supplementary Fig. 1a). On the other hand, p53 protein is known to be frequently dysregulated in cancer compared to health controls. Indeed, our co-expression analysis using anti-TUSC3 and anti-p53 antibodies showed there was an inverse correlation in only metastasized lung cancer patient samples (Fig. 4j and Supplementary Fig. 8j). These observations suggest that the other factors could be also involved in the regulation of p53 in TUSC3 deficiency as well as our observation of HRD1-dependent ERAD. Moreover, it has been well characterized that p53 protein is regulated by another several factors including hypoxia and another E3 ubiquitin ligase, MDM2 protein²⁰. Therefore, we believe that TUSC3 is likely to regulate cancer metastasis rather than a general tumor suppressor in, at least, lung cancer context. What other factors regulate p53 protein to mediate TUSC3-dependent tumor suppressive roles in a certain context could be an interesting another project.

** Please note that we changed the figure format for Supplementary Fig. 1a because there was no any information about the expression on y-axis in previous version.

Previous Supplementary Fig. 1a

Revised Supplementary Fig. 1a

3. Regarding the CRISPR knockout experiments, how to rule out off-target effects?

Response) As the reviewer pointed out, we also agree the possibility of off-target effect in CRISPR KO experiment because this is a kind of general concerns regarding CRISPR technology. However, the concern in this study could be ruled out by the following observations. First, we consistently showed similar effect of the KO cells in shRNA-mediated TUSC3 knock-down cells (Fig. 2a,d, Fig. 5d,f and Supplementary Fig. 3d, Supplementary Fig. 9a). Second, some results with the KO cells were rescued by TUSC3 overexpression. Additionally, we also showed functionally consistent results using some rescue experiments for the target genes in TUSC3 deficiency such as HRD1 and p53. Moreover, the overexpression experiment with TUSC3 plasmid showed consistent results (Fig. 3b-j, Fig. 4f,g,h,I and Supplementary Fig. 3a,b,e). Third, transient suppression of TUSC3 gene enforced by siTUSC3 siRNAs also showed consistent effects with TUSC3 KO cells in the suppression of metastatic potential (Fig. 2e, 5e and Supplementary Fig. 3g-i). Therefore, we believe our observations using the knock-out constructs would be direct effects by TUSC3 and/or HRD1 KO.

4. The sequences of shRNA, siRNA, and gRNA need to be provided. Also, the DNA sequencing result of the CRISPR knockout clones should be shown.

Response) In response to the reviewer's suggestion, we added all the sequence information used in this paper including shRNAs, siRNAs and sgRNAs to Supplementary Table 7. Moreover, we added the changed genomic DNA sequences in the representative TUSC3KO and TUSC3/HRD DKO cells used for functional studies (Supplementary Fig. 2h,i or Fig. 8 for Reviewer). Additionally, the primer sequences to analyze the genomic DNA sequences were also shown in Supplementary Table 7. Regarding this, we revised Supplementary Table 7 and added new Supplementary Fig. 2h and 2i.

Fig. 8 for Reviewer showing that the changed genomic DNA sequences in the TUSC3 (a) or HRD1 (b) KO cells.

5. Figure 4i: the arrows are too big and it is hard to see the lung nodules.

Response) We thank you for the detailed comment. We changed the figures into new one.

Previous Fig. 4i

Revised Fig. 4i

Reviewer #3 (Remarks to the Author):

The revised version of the manuscript has responded to the comments by reviewer 3 and resolved most of the issues. The manuscript was improved and acceptable for publication.

Response) We would appreciate it.

References for Responses

- 1 Yoshida, H., Matsui, T., Yamamoto, A., Okada, T. & Mori, K. XBP1 mRNA is induced by ATF6 and spliced by IRE1 in response to ER stress to produce a highly active transcription factor. *Cell* **107**, 881-891 (2001).
- 2 Clarke, H. J., Chambers, J. E., Liniker, E. & Marciniak, S. J. Endoplasmic reticulum stress in malignancy. *Cancer cell* **25**, 563-573, doi:10.1016/j.ccr.2014.03.015 (2014).
- 3 Fang, S. *et al.* The tumor autocrine motility factor receptor, gp78, is a ubiquitin protein ligase implicated in degradation from the endoplasmic reticulum. *Proceedings of the National Academy of Sciences of the United States of America* **98**, 14422-14427, doi:10.1073/pnas.251401598 (2001).
- 4 Shmueli, A., Tsai, Y. C., Yang, M., Braun, M. A. & Weissman, A. M. Targeting of gp78 for ubiquitin-mediated proteasomal degradation by Hrd1: cross-talk between E3s in the endoplasmic reticulum. *Biochemical and biophysical research communications* **390**, 758-762, doi:10.1016/j.bbrc.2009.10.045 (2009).

- 5 Xu, Y. M. *et al.* HRD1 suppresses the growth and metastasis of breast cancer cells by promoting IGF-1R degradation. *Oncotarget* **6**, 42854-42867, doi:10.18632/oncotarget.5733 (2015).
- 6 Qu, L. *et al.* Endoplasmic reticulum stress induces p53 cytoplasmic localization and prevents p53-dependent apoptosis by a pathway involving glycogen synthase kinase-3beta. *Genes & development* **18**, 261-277, doi:10.1101/gad.1165804 (2004).
- 7 Yamasaki, S. *et al.* Cytoplasmic destruction of p53 by the endoplasmic reticulum-resident ubiquitin ligase 'Synoviolin'. *The EMBO journal* **26**, 113-122, doi:10.1038/sj.emboj.7601490 (2007).
- 8 Guervos, M. A. *et al.* Deletions of N33, STK11 and TP53 are involved in the development of lymph node metastasis in larynx and pharynx carcinomas. *Cellular oncology : the official journal of the International Society for Cellular Oncology* **29**, 327-334 (2007).
- 9 Stevenson, J., Huang, E. Y. & Olzmann, J. A. Endoplasmic Reticulum-Associated Degradation and Lipid Homeostasis. *Annual review of nutrition* **36**, 511-542, doi:10.1146/annurev-nutr-071715-051030 (2016).
- 10 Cai, Y. *et al.* Loss of Chromosome 8p Governs Tumor Progression and Drug Response by Altering Lipid Metabolism. *Cancer cell* **29**, 751-766, doi:10.1016/j.ccell.2016.04.003 (2016).
- 11 Bova, G. S. *et al.* Homozygous deletion and frequent allelic loss of chromosome 8p22 loci in human prostate cancer. *Cancer research* **53**, 3869-3873 (1993).
- 12 Zemliakova, V. V. *et al.* [Profile of methylation of certain tumor growth suppressing genes in non-small cell lung cancer]. *Molekuliarnaia biologii* **37**, 983-988 (2003).
- 13 Duppel, U., Woenckhaus, M., Schulz, C., Merk, J. & Dietmaier, W. Quantitative detection of TUSC3 promoter methylation - a potential biomarker for prognosis in lung cancer. *Oncology letters* **12**, 3004-3012, doi:10.3892/ol.2016.4927 (2016).
- 14 Ferreira, H. J. *et al.* Circular RNA CpG island hypermethylation-associated silencing in human cancer. *Oncotarget* **9**, 29208-29219, doi:10.18632/oncotarget.25673 (2018).
- 15 Ahrendt, S. A. *et al.* p53 mutations and survival in stage I non-small-cell lung cancer: results of a prospective study. *Journal of the National Cancer Institute* **95**, 961-970 (2003).
- 16 Comprehensive genomic characterization of squamous cell lung cancers. *Nature* **489**, 519-525, doi:10.1038/nature11404 (2012).
- 17 Gu, J. *et al.* TP53 mutation is associated with a poor clinical outcome for non-small cell lung cancer: Evidence from a meta-analysis. *Molecular and clinical oncology* **5**, 705-713, doi:10.3892/mco.2016.1057 (2016).
- 18 Fan, X. *et al.* Decreased TUSC3 Promotes Pancreatic Cancer Proliferation, Invasion and Metastasis. *PloS one* **11**, e0149028, doi:10.1371/journal.pone.0149028 (2016).
- 19 Horak, P. *et al.* TUSC3 loss alters the ER stress response and accelerates prostate cancer growth in vivo. *Scientific reports* **4**, 3739, doi:10.1038/srep03739 (2014).
- 20 Wade, M., Li, Y. C. & Wahl, G. M. MDM2, MDMX and p53 in oncogenesis and cancer therapy. *Nature reviews. Cancer* **13**, 83-96, doi:10.1038/nrc3430 (2013).

Reviewers' Comments:

Reviewer #1:

Remarks to the Author:

In this revised manuscript the authors have addressed all our concerns and perform the requested complementary experiments. The manuscript was greatly improved and it is now acceptable for publication.

Reviewer #2:

Remarks to the Author:

The authors addressed all my remaining comments adequately (regarding point #4, the DNA sequencing results of CRISPR KO clones should be included in the paper). I have no additional criticisms.

REVIEWERS' COMMENTS:

Reviewer #1 (Remarks to the Author): In this revised manuscript the authors have addressed all our concerns and perform the requested complementary experiments. The manuscript was greatly improved and it is now acceptable for publication.

Reviewer #2 (Remarks to the Author): The authors addressed all my remaining comments adequately (regarding point #4, the DNA sequencing results of CRISPR KO clones should be included in the paper). I have no additional criticisms. Response) We are uncertain how to provide the files for a bunch of the Sanger sequencing results. We are willing to provide it if the editor have any suggestion for this.